# Habitat suitability mapping and landscape connectivity analysis to predict African swine fever spread in wild boar populations: A focus on Northern Italy

Giulia Faustini[1]*, Marie Soret[2,3,4], Alexandre Defossez[2,3], Jaime Bosch[5,6], Annamaria Conte[7], Annelise Tran[3,8]

1 Department of Animal Medicine, Production and Health, University of Padova, Legnaro, Italy, 2 National Research Institute for Agriculture, Food and Environment (INRAE), Montpellier, France, 3 TETIS, Université de Montpellier, AgroParisTech, CIRAD, INRAE, Montpellier, France, 4 Oïkolab, TerrOïko, Sorèze, France, 5 VISAVET Health Surveillance Center, Complutense University of Madrid, Madrid, Spain, 6 Department of Animal Health, Faculty of Veterinary, Complutense University of Madrid, Madrid, Spain, 7 Istituto Zooprofilattico Sperimentale dell'Abruzzo e del Molise 'G. Caporale' (IZS-Teramo), Teramo, Italy, 8 French Agricultural Research Centre for International Development (CIRAD), UMR TETIS, Montpellier, France

* giulia.faustini.1@phd.unipd.it, database.gfa@gmail.com

**Data Availability Statement:** Main code scripts used in the workflow are available at https://doi.org/10.5281/zenodo.14639171. Link to Gitlab

## Abstract

African swine fever (ASF) is a highly contagious disease affecting wild and domestic pigs, characterised by severe haemorrhagic symptoms and high mortality rates. Originally confined to Sub-Saharan Africa, ASF virus genotype II has spread to Europe since 2014, mainly affecting Eastern Europe, and progressing through wild boar migrations and human action. In January 2022, the first case of ASF, due to genotype II, was reported in North-western Italy, in a wild boar carcass. Thereafter, numerous positive wild boars were identified, indicating an expanding wild epidemic, severely threatening Italian pig farming and trade. This study focused on the mapping of the suitable habitats for wild boars and their potential dispersal corridors in Northern Italy, using species distribution models and landscape connectivity analysis. The resulting maps identified areas with higher likelihood of wild boar presence, highlighting their preferential pathways crossing Northern Italy. The distribution of ASF positive wild boars along the major corridors predicted by the model suggests the obtained maps as valuable support to decision-makers to improve ASF surveillance and carcass early detection, aiming for eradication. The applied framework can be easily replicated in other regions and countries.

## Introduction

African swine fever (ASF) is one of the most devastating diseases of swine, caused by a large, enveloped, double-stranded DNA virus (generally referred as ASFV), which is the only member of the *Asfarviridae* family, genus *Asfivirus* [1]. Over the years, ASF has evolved from a localised disease restricted in Sub-Saharan Africa [2], to a major threat to global pig

repository: https://gitlab.com/giuliafaustini1/habitat-suitability-mapping-and-landscape-connectivity-analysisto-predict-african-swine-fever-spread-in-wild-boar-population. Suitability maps, connectivity map and binary risk maps are available at https://doi.org/10.5281/zenodo.14637229.

**Funding:** The research project was made possible by the financial support provided to G.F. by the Società Italiana di Patologia ed Allevamento dei Suini (SIPAS, https://www.sipas.org/chi-siamo/) and by the Erasmus + for traineeship mobility program (https://erasmus-plus.ec.europa.eu/opportunities/opportunities-for-individuals/students/traineeships-abroad-for-students). The funders had no role in study design, data collection and analysis, decision to publish, or preparation of the manuscript. There was no additional external funding received for this study."

**Competing interests:** The authors have declared that no competing interests exist.

populations [3]. Because of its significant health and socioeconomic impact [4], and the importance of early intervention to control new outbreaks, ASF is listed among the notifiable diseases (i.e., disease of public health importance, mandatory to be reported to related authorities) by the World Organization for Animal Health (WOAH).

ASF clinical manifestations may vary from a severe haemorrhagic disease, with hyperacute clinical signs and nearly 100% mortality, to asymptomatic and chronic forms, which turn animals into silent virus carriers contributing to the persistence and dissemination of the disease [5]. The lack of clinical signs is more frequent in African wild suids such as warthogs (genus *Phacochoerus*), bush pigs (*Potamochoerus porcus* and *P. larvatus*) and giant forest hogs (*Hylochoerus meinertzhageni*) [6], while domestic pigs and wild boars more frequently develop the haemorrhagic form. Although seroconversion is observed in the few animals that survive the disease [7–9], the neutralising activity of the antibody response is highly uncertain, making future reinfections possible [10–12]. ASFV is a highly contagious and stable pathogen, able to persist from weeks to months in infected materials (e.g., blood, excretions, tissues), and even more than a year in contaminated animal products (e.g., frozen meat, cured ham) [5]. Transmission and spread of the virus occur mainly through direct contact with infected animals or carcasses, or indirectly through contaminated food, water, and semen, as well as people (e.g., hunters, workers in pig sectors, veterinarians, etc.) or fomites (i.e., any inanimate object/material which is contaminated, carrying and spreading the pathogen) acting as mechanical vectors. The transmission cycle of ASFV can also be sustained by the presence of soft ticks of the genus *Ornithodoros*, in which the virus can persist for more than 5 years [13]. Infection through *Ornithodoros moubata* plays an important role in the sylvatic cycle in Africa [14]. In contrast, in the current European epidemiological situation, despite the presence of some *Ornithodoros* species (e.g., *O. erraticus*, *O. maritimus*), their involvement appears marginal [5].

ASFV exhibits high genetic and antigenic variability, leading to the emergence of 24 known genotypes, five of which adapted to members of the *Suidae* family, without zoonotic potential [1, 15]. Only ASFV genotypes I and II are currently present in European countries, with two distinct epidemiological scenarios. Genotype I, introduced during a first wave in several European countries in the second half of the 1900s [16], is now circulating only in Sardinia, an Italian Island in the Mediterranean Sea. This region is characterised by a peculiar situation of endemicity, maintained by repeated interchanges between free-ranging domestic pigs and wild boars [17]. Genotype II is currently affecting continental Europe since its introduction in Georgia in 2007 via contaminated waste from ships [18, 19]. The virus began its spread to Eastern Europe in 2014, with Lithuania recording the first cases. From there, ASFV genotype II quickly expanded to neighbouring countries, particularly Poland, Latvia, and Estonia, often emerging near their borders with Belarus and the Russian Federation. By 2019, the ASFV genotype II epidemic had spread like wildfire, affecting most of the Baltic states (Lithuania, Latvia, and Estonia). The virus continued its westward expansion, reaching Germany through Poland by late 2019 [18–20]. Notably, in the Baltic states and Poland, the virus has shown remarkable persistence within wild boar populations, with self-sustaining cycles over several years, while causing minimal outbreaks among domestic pigs [19–21]. On a separate front, ASFV genotype II has also reached Romania, following cases recorded in wild boars in neighbouring countries (i.e., Moldova and Ukraine) [22].

While wild boar movements and territorial wild boar density played a key role in local spread [19, 21, 23], long-distance jumps often resulted from human activities. For instance, ASFV genotype II occurred into European countries such as Czech Republic (2017) [24], and Belgium (2018) [25], appearing as isolated cases, likely due to improper disposal of contaminated food by humans [20]. Less clear is the origin of first outbreaks reported in Hungary, Bulgaria, Slovakia and Serbia in both domestic pigs and wild boars [19].

African swine fever (ASF) spread in wild boar populations of Eastern Europe has been governed by host factors (population structure), viral characteristics (strain virulence), and environmental conditions (geographical barriers) [26, 27], with many dynamics still poorly understood [28]. Network analysis in affected European countries showed infection velocities of 2.9–11.7 km/year, excluding human-mediated transmission, with seasonal acceleration during summer [19]. This slow but persistent spread has caused substantial wild boar population declines in Eastern Europe [29, 30] while disrupting regional livestock economics through reduced pigmeat production and exports [31].

The continued geographic expansion of ASFV genotype II reached Italy in 2022, following an unexpected introduction likely human-mediated [32]. While ASFV genotype II incursion was expected from the eastern side, in January 2022, the National Reference Center for the Study of Pestivirus and Asfivirus Diseases (CEREP) at the Istituto Zooprofilattico Sperimentale dell'Umbria e delle Marche (IZSUM) confirmed the first case of ASFV genotype II in Italy, in the Northwest, in the municipality of Ovada, Piedmont [32–34]. Following this first finding, several carcasses positive to ASFV genotype II were detected in Liguria. Over a year later, ASFV genotype II-positive wild boars were found in other regions such as Lombardy, and Emilia-Romagna (Fig 1). So far, in Northern Italy, only nine genotype II outbreaks have been reported in domestic pigs, all in Lombardy region (Italian national epidemiological bulletin, https://storymaps.arcgis.com/stories/9fe6aa3980ca438cb9c7e8d656358f35, accessed on 10[th] July 2024).

Despite the detection of ASFV genotype II in isolated outbreaks in regions of Central and Southern Italy, and in Sardinia [33, 35], the epidemic front currently expanding is the one affecting wild boars in Northern Italy, severely threatening the socio-economic system related to Italian pig husbandry. Northern regions are indeed where most Italian intensive pig farming is concentrated (https://www.istat.it/), primarily aimed at the production of Italian cured ham (e.g., Parma ham). Any further involvement of domestic swine facilities in the ASFV genotype II epidemic due to negligence in biosecurity measures and transmission from the external environment would escalate to deaths or culling of large numbers of animals, as well as bans on the pork trade.

The persistence and spread of ASFV genotype II in Eastern Europe, strongly driven by the wild boar population, along with its eradication through effective hunting campaigns and early detection of carcasses, highlight the crucial need to better understand the distribution and mobility of the wild boar population to halt or at least mitigate the dispersal of the disease [19, 24, 25]. ASFV genotype II spread may vary greatly among the different habitats found in Northern Italy (Alps, pre-Alps, plains) [34, 36]. Wild boar suitable habitats and distribution have been largely estimated at European scale, through the application of different models, cumulatively named as species distribution models (SDM) [37]. SDM are based on the use of species occurrence/sighting data, which are related to explanatory variables describing the environments, to infer the related likelihood of presence [38]. In literature, presence/absence data for wild boar differs considerably, although often in relation with a common/similar set of environmental variables describing topography, climate, human disturbance and land cover [39–46]. Briefly, SDM can be applied with presence-only data (e.g. BIOCLIM model) [41], presence-background data (e.g. MaxEnt model) [43, 47], presence-absence data (e.g. Random Forest model) [41], or following a Bayesian framework (e.g. INLA model) [44]. Since the best modelling technique remains a topic of debate in the scientific community [48–50], a common and accepted approach is to choose the model by comparing performance metrics of multiple models [51–53]. This approach has been suggested also by ENET wild consortium (i.e. an international network of wildlife professionals supported by the European Food Safety Agency) for wild boar abundance predictions to mitigate model-specific biases and potential artifacts which may otherwise impact estimations [37].

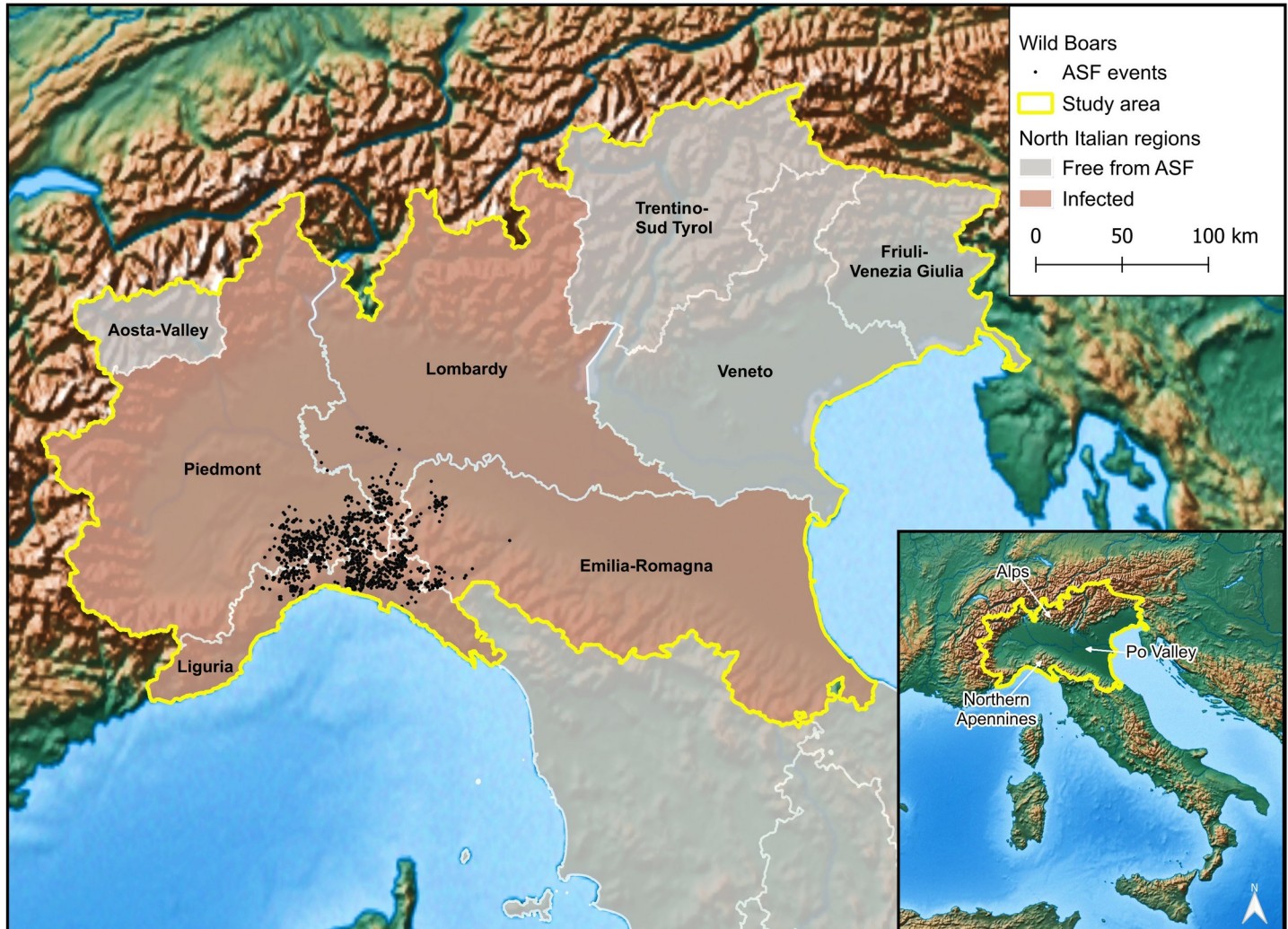

**Fig 1. Map of Northern Italy, showing regions affected by African swine fever virus (ASFV) genotype II on 18th April 2024.** ASFV genotype II outbreak point locations in wild boars are also displayed (World Animal Health Information System (WAHIS)). Base map made with Natural Earth. North Italian regions based on the 2024 regional administrative boundaries from the Italian National Institute of Statistics (https://www.istat.it/).

At European level, a helpful overview on high-risk routes for wild boar migration corridors and ASF spreading is currently available [54]. However, to effectively act at the local level, public authorities would benefit from detailed maps of hotspots for wild boar presence and movements. These maps would allow to target hunting and ASF surveillance activities, such as carcass detection. To fill this knowledge gap, the present study aimed to map suitable habitats for wild boars and their potential corridors of dispersal at high resolution in Northern Italy.

## Material and methods

### Data

**Study area.** The study area included the geographical region of Northern Italy, comprising the following administrative regions: Aosta-Valley, Piedmont, Lombardy, Trentino-Sud Tyrol, Veneto, Friuli-Venezia Giulia, Liguria and Emilia-Romagna (Fig 1). Geographically, Northern Italy features a diverse landscape. The region is bordered by vast mountainous areas

and diverse orography: the Alps to the north and west, and the Northern Apennines to the south. In between lies the Po Valley (Fig 1), the main Italian plain, traversed by the Po River, the longest river in the country. Northern Italy is a densely populated area, with a total surface area of 120,325.67 km$^2$, which concentrates 46.6% (27,490,042 inhabitants out of a total of 58,989,749 inhabitants) of the Italian population (June 5th, 2024, https://www.istat.it/). From a production perspective, the Po Valley is highly industrialised, characterised by intensive agriculture and livestock farming, as well as numerous urban centres of significant industrial and tourist importance.

**Wild boar presence data.** Wild boar presence data largely included occurrences available on https://www.gbif.org, the Global Biodiversity Information Facility (accessed on 15th February 2024, GBIF Occurrence Download https://doi.org/10.15468/dl.cgjtsj). GBIF is a free, open-access database providing data about different taxa, collected by many institutions and associations during surveys and/or citizen science programs. GBIF records located in the study area were selected and processed following the pipeline described by GBIF portal (https://docs.gbif.org/course-data-use/en/data-processing-pipeline.html). In addition, available field records from Veneto and Lombardy, obtained from official routine culling campaigns, aimed at controlling wild boar population expansion, were included in the study [55].

Only records with a known coordinate uncertainty of less than 100 m were considered. The available occurrences were also filtered for the time period 2014–2024, because of the limited data available in previous years. To meet the assumption of independence of records, reduce sampling bias and point clustering, preventing pseudoreplication (samples taken from the same experimental unit are treated as independent replicates) and spatial autocorrelation issues (neighbouring coordinates tend to have similar values for environmental variables) [56–59], the density of occurrences was reduced to a minimum distance of 100 m. Duplicated records were also removed. Data cleaning and processing were performed using R statistical software [60].

**Environmental variables.** Based on models applied in previous studies, and accounting for wild boar ecology, a variety of environmental predictors were selected to depict different aspects of the habitat, such as topography (altitude, slope, topological aspect, topological diversity, topological position), climate (precipitation, temperature), human disturbance (population density, distance from highways, distance from urban areas, road density), land cover and vegetation (Normalized Difference Vegetation Index (NDVI), bare coverage, herbaceous coverage, tree coverage (all intended as percentage 0–100 per cell), distance from crop areas, distance from forest, distance from lakes, distance from rivers, distance from Natura 2000 parks) [37, 61]. Variable description, sources, and processing details are reported in S2 Table. Briefly, all variables were obtained from public databases, reprojected on the same coordinate reference system (RDN2008 / Italy zone (N-E)), and standardised as rasters with a pixel size of 100 m x 100 m (bilinear resampling method), over an extent equal to the study area. Categorical variables (land cover classes) and vector layers (showing lakes, rivers, parks, highways, and roads) were converted to continuous variables, considering the distance from the described feature, or the feature density. Variables with a temporal dimension (NDVI, temperature, precipitation), were processed to account for seasonal variations in habitat use and resource availability for wild boars. Specifically, a single spatial layer was obtained by calculating over the years (period 2014–2023, depending on available data) the overall mean (overall scenario), and the seasonal mean (seasonal scenarios). This approach was chosen to provide a comprehensive view of environmental conditions over time, aligning with the time frame of occurrence data, while simultaneously smoothing out short-term fluctuations and variability [41, 42, 62].

Based on the meteorological convention for the Northern hemisphere, the considered months for seasonal means were December, January, February for winter; March, April, May

for spring; June, July, August for summer; September, October, November for autumn [63]. Spatial layer processing was performed on QGIS Geographic Information System [64] and Python [65].

To reduce collinearity between environmental predictors [66], the selected variables were tested for autocorrelation, via a correlation tree (or cluster dendrogram) obtained with the *stats* library in R [60]. In the obtained correlation tree, a minimum cut-off of 0.5 was considered for variable selection. Based on ecological and spatial criteria, the original resolution, and interpretive ease, only one variable was selected in each cluster of variables with node < 0.5. The selected variables were further tested for multicollinearity with *HH* library [67], calculating the variance inflation factor (VIF) [68], and selecting variables with a maximum value of 5 [43]. Collinearity and multicollinearity assessment was conducted for the full set of variables for the overall analysis, while for seasonal analysis the variables with a temporal dimension covered only the relative months.

The variable included in the study for all scenarios were 13 (S1 Table). Temperature, NDVI, herbaceous coverage, distance from urban areas, and topological diversity were excluded in all scenarios. Precipitation variable was excluded in the overall and summer scenarios; slope variable was excluded only in the summer scenario; tree coverage was excluded in the winter, spring and autumn scenarios, while distance from crop areas was excluded in all scenarios except for summer (S1 Table).

## Modelling workflow

To capture broad, long-term trends as well as seasonal variations in wild boar habitat use and resource availability, both overall and seasonal suitability were modelled. The modelling workflow was structured in two main phases (Fig 2). Firstly, the wild boar occurrence records, and the set of environmental predictors for wild boar distribution, were used as input to model the potential distribution of wild boars in the study area (habitat suitability) for the overall and seasonal scenarios using SDM (Fig 2A) [38, 69]. Then, the obtained overall suitability map was used to perform the landscape connectivity analysis of the study area (Fig 2B).

**Species distribution modelling (SDM).** For the overall scenario, all the occurrences were included in the suitability modelling, using an overall average for variables with time dimension. For the seasonal scenarios, only the related monthly records were used, with mean seasonal layer for precipitation variable.

Since for wild boars, real absence data are hardly available, SDM techniques rely on a sample of points from the study area, called "pseudoabsences" (PsA), which are used to sample the background, providing information about the environmental factors potentially driving the distribution of presence records [70]. Occurrences available on naturalistic databases (such as GBIF) are often spatially biased toward areas easily accessible by humans [70]. If PsA are randomly generated, the final model will show that the distribution of the species correlates higher with human facilities (e.g., roads, towns), rather than with real driving factors [70]. To reduce the sample selection bias and target the modelling on species distribution, rather than on survey effort, PsA were selected using the same underlying bias as occurrence data [71, 72]. With this approach, the model will capture any differentiation between the distribution of presence records and that of PsA, better highlighting preferential habitats, rather than area accessibility. A kernel density surface with higher probabilities in areas with more presence data was used as sampling bias surface, to generate a number of PsA equal to the number of occurrences data [73, 74].

The habitat suitability modelling was performed using the algorithms available in the package *Biomod2* in R [75], with default options: Artificial Neural Network (ANN), Classification

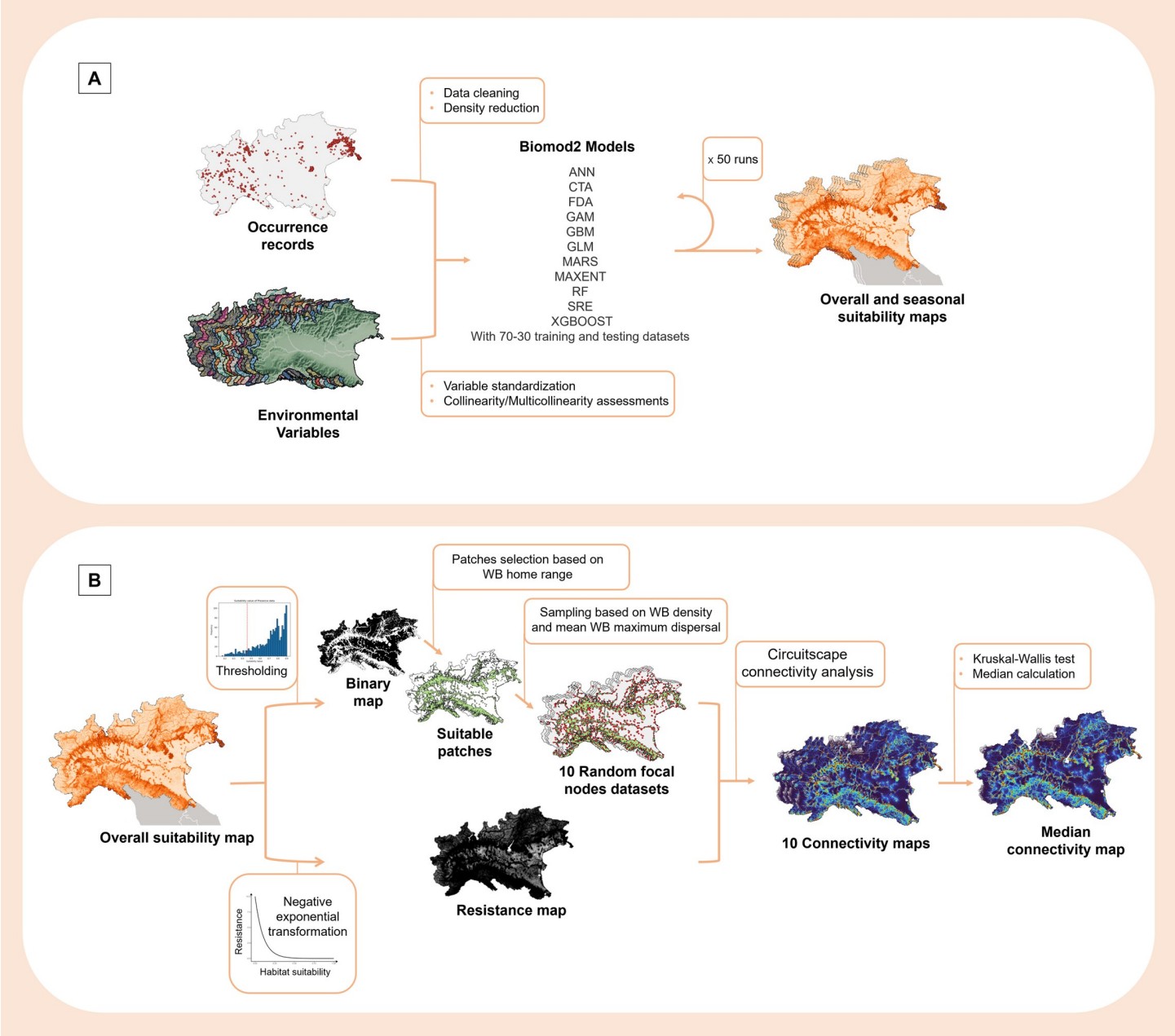

**Fig 2. Habitat suitability mapping and landscape connectivity analysis workflows.** (A) Modelling workflow to predict suitable habitats for wild boar presence from occurrence records and environmental predictors, (B) and to estimate the landscape connectivity of the study area considering the most suitable areas. "WB" is used as abbreviation of "wild boar". Environmental variables made with Natural Earth for illustrative purpose only. Base map of Italy in the suitability maps based on the 2024 regional administrative boundaries from the Italian National Institute of Statistics (https://www.istat.it/).

Tree Analysis (CTA), Flexible Discriminant Analysis (FDA), Generalized Additive Model (GAM, or BAM), Gradient Boosted Machine (GBM), Generalized Linear Model (GLM), Multiple Adaptive Regression Splines (MARS), Maximum Entropy (MAXNET), Random Forest (RF), Surface Range Envelop / BIOCLIM (SRE), eXtreme Gradient Boosting Training (XGBOOST) (Fig 2A). The protocol consisted of 50 runs per algorithm, with the same weight for presence and PsA data, using 70% of the datasets for training and 30% for testing [74],

splitting randomly the data between the train and test datasets. The best algorithm was chosen considering True Skills Statistics (TSS) and AUC (Area Under ROC (Relative Operating Characteristic) Curve) as evaluation metrics, and excluding the algorithms displaying overfitting. For the selected algorithm, an ensemble model was computed through the median of all runs with AUC greater than 0.8 [76], and further evaluated with AUC, TSS and Cohen's Kappa (Kappa) [77–79]. A final continuous prediction raster of habitat suitability (suitability map ranging from 0 to 1) was obtained forecasting the ensemble model.

The presence of a significant difference between the suitability maps under the different scenarios was tested with Kruskal-Wallis test, while the pairwise multiple-comparison was performed through the Dunn test. The distribution of the overall suitability value of the presence data included in the model was compared through a Mann-Whitney test with the one of the unused records, removed during the occurrence's density reduction.

**Connectivity analysis.**   Since carcass search and removal is a process that should ideally be conducted routinely in the restriction zones (i.e., infected areas or surrounding areas under surveillance measures), especially in the epidemic phase [80], connectivity was estimated only for the overall scenario. The landscape connectivity of the study area was modelled on the overall suitability map using Circuitscape, an open-source program implemented in Julia [81]. Circuitscape is based on circuit theory, and can operate on networks of nodes or raster grids. With raster data, Circuitscape converts raster grids into electrical networks, in which each cell becomes a node that is connected to the neighbouring cells by resistors. According to settings, from a current source, Circuitscape estimates the flow between focal nodes, simulating animal movements. Input files in this study included a focal nodes location file, a resistance map, and a raster mask file. All input files were obtained using QGIS and Python.

Due to high computational and storage demands of Circuitscape, focal nodes were obtained using the following workflow (Fig 2B). The continuous overall suitability prediction was converted into a binary map considering as threshold the $10^{th}$ percentile of the suitability score distribution of presence data, assuming that areas with higher suitability score better reflect wild boars' habitat. This criterion has been largely adopted in the ecological field, showing good performances in obtaining suitable patches [74, 82–84]. The regions with suitability scores greater than the threshold (suitable areas) illustrated in the binary map, were converted into vector patches, excluding those smaller than 4.4 km$^2$, the mean population home range size for wild boars [54, 85, 86]. Ten sets of random points were generated within the obtained patches, taking into account wild boar animal density within patches [42], and a minimum distance of 12 km, the mean population maximum dispersal for wild boars (Fig 2B) [86]. Once converted to raster format, the obtained random points served as focal nodes.

The resistance map conceptually represents the opposite of habitat suitability: low suitability/permeability areas correspond to higher resistance, and vice versa. The resistance map was computed applying a negative exponential function, such that *resistance = e[(ln(0.001)÷threshold)×HS]×10$^3$*, which enhances the barrier effect of less-suitable sites, returning smoother effects [87–90].

To exclude cells outside the considered Italian regions, and big lakes from the analysis, a raster mask file was created, setting the corresponding pixels as NODATA values. Thus, NODATA values were excluded from the resistance map, and treated like barriers [81].

The connectivity analysis was conducted for all 10 sets of random points (focal nodes) using the "one-to-all" mode, which iterates across all random points, such that in each new iteration a different random point acts as the current source. Julia 1.10.2 version was used to run Circuitscape program [81].

To test if the 10 connectivity maps estimated were comparable, the presence of a significant difference in terms of connectivity values between maps was evaluated by Kruskal-Wallis test

(Fig 2B). Since this test compares the median values of the maps (i.e., non-parametric test), a final connectivity map was obtained calculating the median of the 10 connectivity maps, reducing also the influence of potential outliers.

## Model evaluation

The geographical coordinates of Italian ASF outbreaks were downloaded from the World Animal Health Information System (WAHIS), the reference database of the World Organisation for Animal Health (WOAH) (accessed on 18[th] April 2024). Outbreaks related to domestic pigs were excluded, and only notifications corresponding to ASF genotype II positive wild boars within the study area were included in the validation of the obtained connectivity map. From this point forward, all references to ASF outbreaks/cases/events/findings pertain specifically to ASFV genotype II.

To test if ASF events were associated with higher connectivity values, the distribution of the connectivity values at ASF event locations was compared with the one of random points in a minimum bounding geometry (convex hull) enclosing ASF outbreaks through Mann-Whitney test. A set of random points equal to the number of events were generated 100 times, without overlapping the location of the ASF cases, assuming that the entire territory in which ASF positive wild boars were found was scanned. Random points and ASF events connectivity values were extracted from the median connectivity map. The proportion of significant p-values was calculated.

For each of the 100 sets of random points, a dataset of ASF events and random points locations and relative connectivity values was created. The ROC curve was analysed for all the 100 datasets, using *pROC* package in R [91]. The average AUC, the lowest threshold corresponding to 0.9 of sensitivity, and the average best closest top left threshold (the point closest to the top-left part of the ROC curve, corresponding to the best compromise between sensitivity and specificity) were calculated. Binary risk maps of the median connectivity maps were obtained applying the calculated thresholds, and their true positive rate (outbreaks correctly classified as carcass location) was calculated. To increase the chance of wild boar detection, the search of wild boars in the infected areas and in those under surveillance are usually more focused in forest-covered areas [80, 92]. To compare the predictive power of the estimated risk maps with that of the land cover classes only, the proportion of correctly classified ASF positive wild boars was also calculated by land cover class (urban, crop, forest, water). The land cover classes were obtained by reclassifying the Corine Land Cover (CLC) dataset produced within the frame of the Copernicus Land Monitoring Service (S2 Table). All the statistical analysis was performed in R.

## Results

### Presence data and environmental variables

A total of 2496 wild boar occurrences were collected for the period 2000–2024: 2266 downloaded from GBIF data portal, while 230 from official routine culling campaigns. After the data cleaning process to reduce the spatial autocorrelation between occurrences, 1306 presence data were finally included in the overall model, covering the period from 2014 to 2024 (Fig 2A). For the seasonal scenarios, the occurrences included were 293 for winter, 312 for spring, 434 for summer, and 451 for autumn.

After standardisation, all variable layers had a dimension of 26,306,996 grid cells. After the screening for the presence of collinearity and multicollinearity described in material and methods section, out of 20 variables originally considered, 13 variables were selected for the overall and seasonal suitability models, with specific differences according to the scenario (S1 Table).

## Suitability maps

Among the algorithms available in *Biomod2* package, only RF, XGBOOST, and GBM showed acceptable performances for all the considered scenarios (overall and seasonal estimations). On average, the mean and the standard deviation (sd) of the evaluation metrics were AUC equal to 1.00 (sd = 0.00), TSS equal to 0.99 (sd = 0.00) for both RF and XGBOOST, and AUC equal to 0.95 (sd = 0.00), TSS equal to 0.77 (sd = 0.02) for GBM (details per scenario in S3 Table). The other algorithms (ANN, CTA, FDA, GAM, GLM, MARS, MAXNET, SRE) resulted in lower performances (average values: ROC < 0.89; TSS < 0.62).

For all considered scenarios, GBM model was finally preferred for ensemble modelling computation, because of its balanced combination of spatial generalization across the study area and predictive performances. Compared to other models, GBM provided continuous suitability estimates across the study area, in association with the strongest predictive power in terms of evaluation metrics, except for RF and XGBOOST, which showed perfect performance metrics (S3 Table) [93–95]. However, RF and XGBOOST were excluded since the excellent results depicted by the metrics may be related more to an overfitting/overlearning of the occurrences used, than actually to perfect predictive performances (S3 Table) [53, 96–99]. The GBM-based ensembled model was evaluated as acceptable for all the considered performance metrics, in all scenarios (Table 1). The mean values obtained were TSS = 0.73 (sd = 0.08), AUC = 0.94 (sd = 0.03), KAPPA = 0.73 (sd = 0.08). All the above evaluation metrics are rounded to the second decimal digit.

The first variable for importance was the distance from the forest in all scenarios, while the second one was always altitude, except for the winter scenario which was more influenced by seasonal precipitation. In particular, according to ensemble model response curves (S1 Appendix), for all scenarios, the habitat suitability decreased sharply with altitude and distance to forest, but slightly with bare coverage, slope, distance from lakes, distance from parks. On the contrary, the suitability mildly increased with tree coverage, distance from river, road density. The correlation between the suitability and the variables "precipitation", "distance from highways", "topological aspect", "topological position", varied according to season and/or range of suitability. The suitability didn't seem to be particularly influenced by the variability in the human population density and distance from crop. See further response curves details by scenario in S1 Appendix.

Visually, all the suitability maps displayed mountainous and hilly areas with good forest cover as highly suitable areas for the presence of wild boar (Fig 3B and Fig B in S1 Fig). With few exceptions, the lowest suitability was described in the area of the Po Valley, especially the lowlands, and in the northernmost Alps. Cells with suitability value greater than 0.6 were

**Table 1. Evaluation metrics of the ensemble Gradient Boosted Machine (GBM) model and suitability value statistics under the different scenarios.**

| Scenario | Evaluation metric | | | Suitability value | | | |
|---|---|---|---|---|---|---|---|
| | TSS[a] | AUC[b] | KAPPA[c] | mean | sd | min | max |
| Overall | 0.60 | 0.89 | 0.60 | 0.38 | 0.19 | 0.12 | 0.91 |
| Winter | 0.81 | 0.97 | 0.81 | 0.33 | 0.21 | 0.07 | 0.93 |
| Spring | 0.76 | 0.95 | 0.75 | 0.40 | 0.19 | 0.10 | 0.89 |
| Summer | 0.75 | 0.95 | 0.74 | 0.35 | 0.20 | 0.08 | 0.92 |
| Autumn | 0.75 | 0.94 | 0.74 | 0.35 | 0.18 | 0.08 | 0.92 |

[a] True Skills Statistics

[b] Area Under ROC (Relative Operating Characteristic) Curve

[c] Cohen's Kappa

mainly located in the pre-Alps (i.e., groups of mountains and highlands from Lake Maggiore till the border with Slovenia, located on the inner side of the Alps and gradually lower towards the plain), the Northern Apennines, central Piedmont at the province of Asti, the Ticino park area, along the Po River in the Po Valley, and the regional parks of Colli Berici and of Colli Euganei in Veneto (Fig 3B, Fig B in S1 Fig and S2 Fig).

The mean suitability value of Northern Italy, under the overall scenario, is 0.38 (0.12–0.91). On average, the season with the highest suitability was spring (0.40, 0.10–0.89), followed by autumn (0.35, 0.08–0.92), summer (0.35, 0.08–0.92), and winter (0.33, 0.07–0.93). Here reported suitability values are rounded to the second decimal digit. A significant difference was observed in the suitability of the study area under the different scenarios. Specifically, all the five suitability maps resulted significantly different among each other at the *post hoc* analysis (p-value < 0.001).

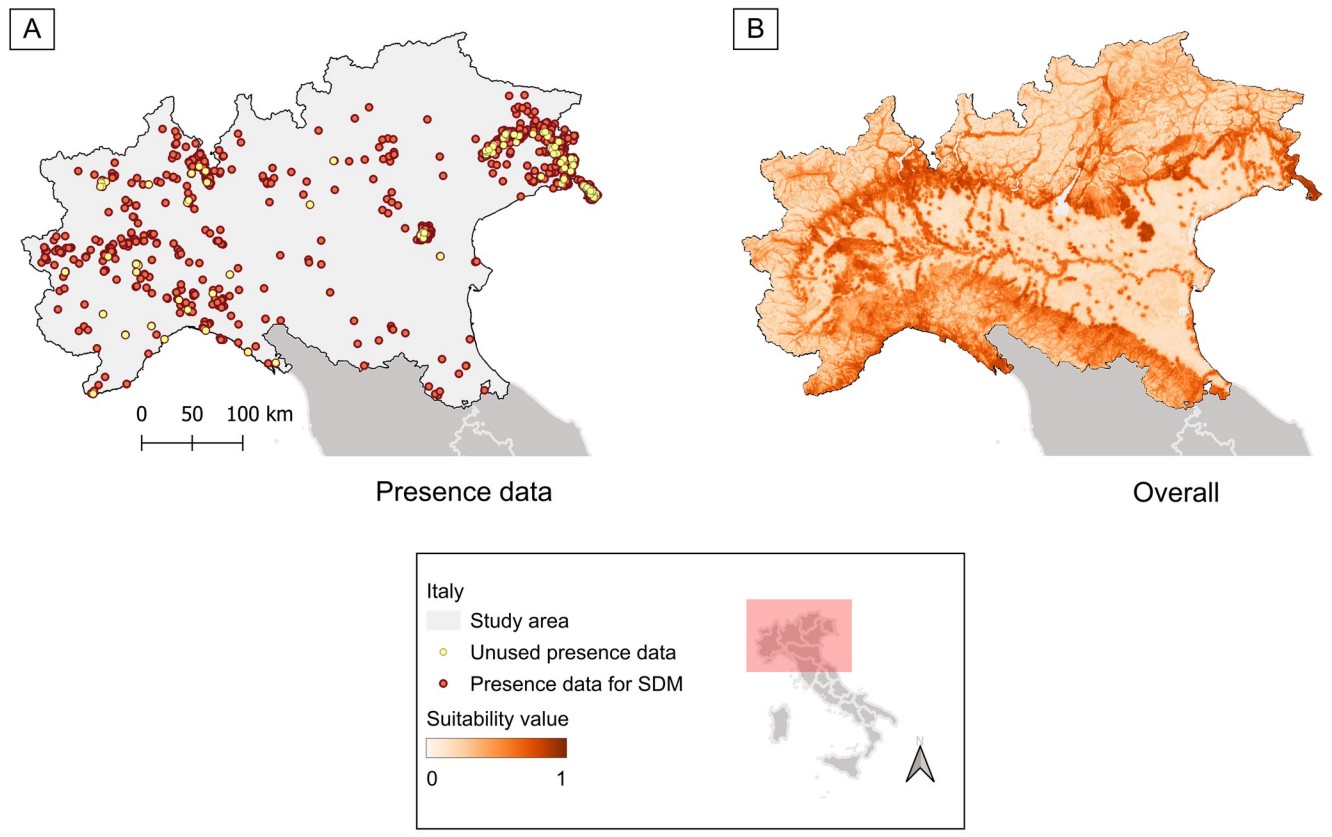

**Fig 3. Distribution of wild boar records and estimated suitability map for the overall scenario.** (A) Distribution of presence data in the study area. Occurrence records included in the final model are displayed as red points, while those removed to meet the assumption of independence are shown as yellow points. (B) Map describing habitat suitability for the presence of wild boars in Northern Italy in an overall scenario. Seasonal suitability maps are available in S1 Fig. Original raster files can be downloaded from S2 File. Base map of Italy based on the 2024 regional administrative boundaries from the Italian National Institute of Statistics (https://www.istat.it/).

Considering the overall scenario, no significant difference was detected between the suitability value distributions of the presence data included in the model and of the unused records, excluded during the density reduction (p-value = 0.36).

### Connectivity map and ASF outbreaks spatial distribution

The threshold corresponding to the 10% percentile of the suitability score distribution for the presence data was 0.45. After excluding the suitable regions smaller than 4.4km$^2$, the final patches covered 32.09% of the total study area. The percentages of occurrences included and excluded from the SDM, falling into the patches were respectively 86.52% and 86.93% [43].

Since the ten connectivity maps obtained from the ten different random points datasets were statistically comparable (p-value ≈ 1), a final connectivity map was calculated through the median formula (Fig 4A). In the median map, the mean connectivity value was 0.53x10$^{-2}$ (0.00–74.83x10$^{-2}$). The connectivity value describes the permeability of a grid cell to wild boar movement.

In total, 1758 wild boar ASFV genotype II notifications were downloaded from WAHIS. The mean distance between all the ASF events was 40.4 km, between consecutive ones was 20.8 km. Since three records fell outside the connectivity map, 1755 ASF events were finally included in the study. ASF event locations were associated on average with higher connectivity values (2.5x10$^{-2}$, 0.00–20.0x10$^{-2}$) than those extracted from random background points (1.5x10$^{-2}$, 0.00–24.9x10$^{-2}$), for all the 100 simulations (p-value < 0.001).

### Risk maps for ASF outbreak according to the connectivity value

The median connectivity map demonstrated a fair/acceptable ability to binary classify ASF event records as empty (random point) or as positive wild boar location. The average AUC for the 100 datasets of random and ASF event points was 0.69 (0.67–0.71). The lowest threshold of connectivity value corresponding to 0.90 of sensitivity considering all datasets was 0.45x10$^{-2}$ (min-max). According to this threshold, the derived binary maps resulted in a true positive rate of 90.02% (1580/1755), which is the percentage of ASF events correctly classified as positive wild boar location (Fig 4B). The average closest top left threshold was 1.14x10$^{-2}$ (min-max), and the corresponding binary map reported a true positive rate of 65.41% (1148/1755) (Fig 4C). The land cover class with the highest true positive rate was "forest" class (55.44%, 973/1755), followed by "crop" (36.98%, 649/1755), "urban" (7.29%, 128/1755), and "water" (0.28%, 5/1755).

## Discussion

This study performed a high-resolution estimation of the suitability of Northern Italy for wild boar presence and of the main dispersal corridors, describing the connectivity map as a potential tool for ASF control and surveillance. The suitability and connectivity maps were obtained by taking into account topographic, climatic, land cover and anthropological aspects that influence wild boar behaviour, outlining and summarising the complexity of wild boar ecology in a single index.

### Northern Italy suitability for wild boars

Using input data mainly from open-source databases, the developed suitability maps visually resembled the pattern already described in previous studies [40, 100], also by season, showing high-resolution estimations (Fig 3B and Fig B in S1 Fig). In addition to the higher resolution of the maps presented here compared to those found in the literature [37, 40, 42, 100, 101], the

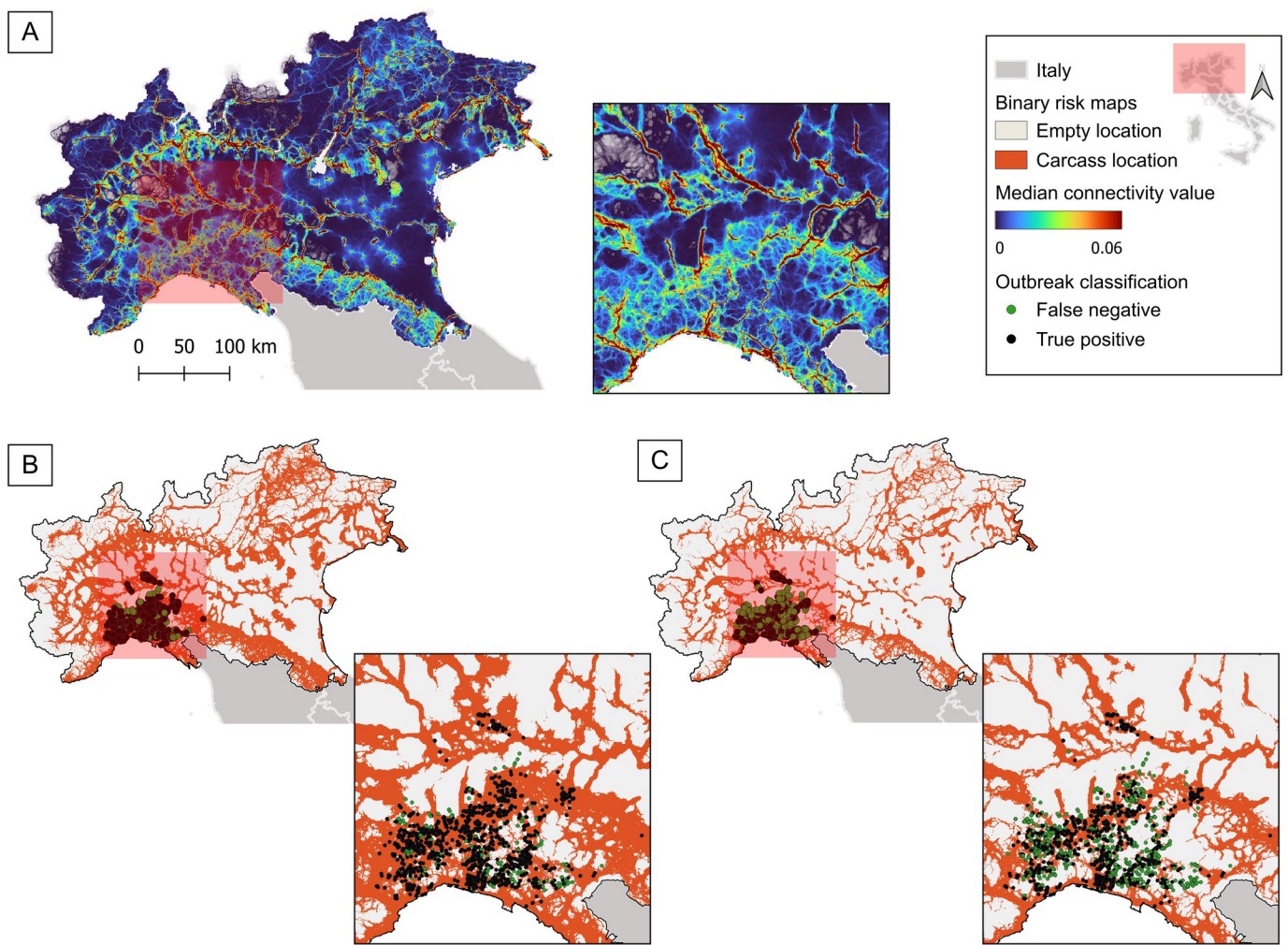

**Fig 4. Connectivity map and binary risk maps according to different threshold.** (A) Map describing landscape connectivity in Northern Italy according to wild boar ecology, highlighting the main corridors of dispersal. The connectivity value describes the permeability of a grid cell to wild boar movement. The scale was resized considering the interval 2–98% of the connectivity value distribution, to highlight major corridors. A zoom on ASF affected area is shown. According to different thresholds of the connectivity value, different binary maps can be obtained: (B) binary risk map with a true positive rate of 90.02%, maximizing the sensitivity of the map in correctly classifying African swine fever (ASF) events due to ASF virus genotype II. A zoom on ASF wild boar cases is shown; (C) binary risk map with a true positive rate of 65.41%, respecting the best compromise between sensitivity and specificity in classifying ASF events due to ASF virus genotype II. A zoom on ASF wild boar cases is shown. Original raster files can be downloaded from S2 File. Base map of Italy in the suitability maps based on the 2024 regional administrative boundaries from the Italian National Institute of Statistics (https://www.istat.it/).

influence of each variable on wild boar presence was also detailed (S1 Appendix). The low average suitability values reported for Northern Italy could be primarily influenced by the large area of the Po Valley, a wide plain poorly suited for wild boar presence, except for suitable "islands" at regional parks, and the strip that traces the path of the Po River. The area of the Northern Apennines and the pre-Alps creates a frame of suitable areas that almost continuously encircles the entire Po Valley (Fig 3B and Fig B in S1 Fig and S2 Fig).

In Northern Italy, wild boars seem to prefer moderate altitudes, below 1000 meters (S1 Appendix), in correspondence with forested areas. The strong anthropization, and the low vegetative cover of the Po Valley could therefore be the limiting factors for wild boar distribution in this area. On the contrary, the more flourishing vegetation present in regional parks, as well as in the area of the Apennines and pre-Alps, could provide food and shelter for wild

boars, favouring their presence (Fig 3B and Fig B in S1 Fig and S2 Fig). Moreover, these areas are characterised by agricultural and commercial activities that interface with or are interspersed among wooded areas (e.g., vineyards and olive trees, orchards, chestnut groves). Such environments could provide advantage for opportunistic and highly adaptable animals like wild boars [102, 103].

Observed seasonal variations, such as increased suitability for wild boar presence in spring, summer, and autumn (Fig B in S1 Fig), may be linked to difference in precipitations and occurrences' density due to reproductive cycles, food availability, and seasonal hunting regulations. For instance, a greater wild boar density in spring may be associated to higher precipitations which are linked to more food resources [102, 104–106]. Higher wild boar activity in spring and summer may correspond to the birth of piglets, increased movement in search of food, and reduced hunting pressure, which typically occurs in autumn and winter [107, 108]. From a management perspective, these findings could inform targeted strategies, such as timing interventions to optimize hunting schedules, disease surveillance plans, and the implementation of biosecurity measures. However, also human-related reporting biases may have shaped the obtained suitability maps [70, 71, 109].

The observed seasonal differences also seem to be only partially supported by the pattern of confirmed cases of ASF in wild boars (Italian national epidemiological bulletin, https:// storymaps.arcgis.com/stories/9fe6aa3980ca438cb9c7e8d656358f35, accessed on 20[th] June 2024), which so far has been upward from October to May, and downward from May to September. Whether this pattern of cases is explained by actual higher suitability in the intermediate seasons (spring and fall), rather than by other factors also accounting for population turnover and latency times between infection and death, needs further investigation [28].

As any model, suitability maps represent a simplification of reality and are the result of the variables and presence data included in the study, each with its limitations and potential biases. A ten-year period was chosen for NDVI and climate variables to emphasize broader landscape trends in wild boar habitat suitability. This approach offered a stable view of habitat conditions that is less affected by short-term fluctuations and more reflective of long-term environmental patterns [41, 42]. However, it may overlook annual fluctuations, especially for NDVI, which can vary due to climate changes or shifts in land use. The exclusion from the SDM of NDVI and temperature variables due to their collinearity with other variables, such as topographic and land cover characteristics, suggests that their variability and influence on wild boar ecology are likely captured by more static habitat features in the considered overall and seasonal time spans. To provide year-specific prediction, each year of wild boar data should be matched with the corresponding yearly environmental variables when available [62, 110]. Future research could incorporate this temporal variability for more precise seasonal or overall predictions of habitat suitability [110].

The greater importance of topographic and land cover variables, besides being compatible with wild boar ecology, was probably influenced by the scale chosen. Indeed, precipitation is a variable whose effect tends to be more measurable and significant at lower resolutions, and less at local scales [111, 112]. Moreover, in this kind of estimation, despite the minimization of the correlation between the chosen variables, the importance of a variable may be influenced by its correlation with real drivers of wild boar distribution in Northern Italy that were instead not included in the model, rather than by its direct contribution. Additionally, limiting the analysis on Northern Italy, unavoidably distort spatial estimations at the edge (i.e., edge effect), because of the exclusion of further occurrences or variable values potentially located right across the border of the study area, which would have provided related information in the model [113, 114]. To address these considerations, future research could explore multi-resolution approaches to

balance local and broader-scale drivers of distribution, potentially enhancing model accuracy and applicability for different ecological and management contexts.

The suitability estimates displayed by the maps were also influenced by the occurrence records available, as they were biased by an unbalanced sampling effort, given by the heterogeneous distribution and activity of data providers (naturalists, researchers, hunters) (Fig 3A and Fig A in S1 Fig) [70, 71, 109].

In Northern Italy, the discontinuous natural parks (S2 Fig), the heterogeneity of the landscape, and the uneven/clustered anthropogenic pressure may have facilitated sampling in areas more easily accessible to humans, affecting the accuracy of estimates in greener, sloping, and border areas (e.g., Northern Apennines, and Alps). In regions featured by more homogeneous landscape and anthropogenic pressure, the estimation of land suitability could return better and more accurate performances in areas where greater wild boar suitability is expected, as smaller differences in accessibility would allow the variability associated with ecological factors to emerge.

Even variations in suitability between seasons were probably dependent more on the different sizes and distribution of occurrences than on the variables included. The variable "precipitation" was in fact the only one with a time dimension included in the final models, and it was among the most important variables only for the winter season. Any explanation of observed seasonal differences thus remains speculative. Increasing the number of occurrence records, collected in a homogeneous, complete, and widespread manner both spatially, over the entire study area, and temporally, could enhance the accuracy of predictions of Northern Italy suitability to the presence of wild boar. Additionally, the incorporation of more variables related to seasonal aspects (e.g., seasonal variability of food resources) is needed to offer a more comprehensive understanding of the wild boar's spatial behaviour across seasons, which in this study remained obscure. A more dynamic approach adopting also daily or weekly variables, instead of static mean, together with VHF (very high frequency) radio-tracking data could validate here presented findings, and further enhance the knowledge on wild boar habitat preferences across diverse landscapes and seasonal conditions [62, 110, 115].

The evaluation of different models allowed the identification of the best-performing one for the specific dataset and objectives of the study. The analysis demonstrated that not all models yield equivalent results. Despite their strong predictive performance, RF and XGBOOST algorithms were not selected for the final analysis. These algorithms exhibited performance metrics approaching or equal to 1 (S3 Table), which, while mathematically optimal, warrant careful interpretation in ecological contexts [53, 96]. Near-perfect metrics could indicate overfitting, suggesting that the models may have captured noise patterns specific to the training data rather than true ecological relationships. Additionally, RF and XGBOOST introduce greater model complexity compared to GBM, which can increase the risk of overfitting and reduce results interpretability [96]. Such overfitting could ultimately limit model transferability and practical application in novel scenarios [53, 97–99]. The minimization of sampling bias through the choice of pseudoabsences as described in [70], the reduction of occurrences density, and the final choice of a model (GBM) with less overfitting behaviour and excellent predictive capabilities even from presence-only data, still resulted in optimal estimation performance for all considered scenarios (Table 1) [93–95]. The obtained suitability maps thus contribute to a deeper understanding of the real distribution of wild boar in the territory. The final suitability maps were an ensemble prediction from multiple runs of an individual algorithm (GBM). Future research could explore the potential benefits of an ensemble modelling combining different algorithms [116, 117], leveraging the strengths of each, to potentially provide more robust estimations on wild boar distribution [37, 49].

## Northern Italy main corridors of wild boar dispersal

The suitability pattern was largely traced from the connectivity map, highlighting the main corridors of dispersal of the species, and thus the potential preferential routes of spread of ASFV genotype II in Northern Italy (Fig 4A). Compared with the suitability map, the connectivity estimation has the advantage of attributing a higher probability of being crossed/reached to areas accessible by multiple pathways, according to circuit theory [81]. This approach returns a continuous estimate of the main dispersal corridors accounting for the complexity of animal movement, showing the connection between the most suitable patches.

The presence of high-traffic corridors (i.e., routes with continuous high connectivity values that facilitate the movement of wild boars) crossing all of Northern Italy, poses a significant threat for the spread of ASFV genotype II to the wild and domestic pig populations. These corridors are found in both the highlands (mainly the Northern Apennines and pre-Alps) and the lowlands (Po valley and regional parks) (Fig 4A and S2 Fig). The high connectivity of these corridors, both in isolated high and low-lying lands and in areas close to urban settlements, increases the risk of contact among wild boars and between wild boar and humans. This heightens the potential for undetected circulation in areas that are more difficult to monitor for wild boar carcasses, as well as for long-distance spread of ASFV due to human activity [18, 20]. Furthermore, the high connectivity observed within regional parks suggests them as potential receptacles for ASFV. This makes regional parks a primary focus for enhanced surveillance and wild boar population density control efforts.

The connection of estimated corridors with currently infected areas (Fig 4) severely highlights the urgent need to improve control, surveillance and biosecurity strategies [118, 119]. Future research could examine the alignment of these wild boar dispersal corridors with existing farm locations to better understand transmission risks and refine prevention strategies.

Wild boar search is usually focused in wooded areas, and with a good vegetation cover, which may be more intuitive by human perspective. However, the capacity of forest land cover class to correctly classify ASF event locations close to a random classification, might suggest that a different approach should be adopted. Conversely, the connectivity map demonstrated a strong ability in accurately identifying ASF event outbreak locations (Fig 4B and 4C). ASFV genotype II positive wild boars tended to locate at high levels of connectivity (compared with random points), and the true positive rate of the binary risk maps was higher than that achieved using only land cover classes. These findings suggest the high potential of the obtained connectivity map as a significant tool for controlling the spread of ASFV genotype II across the territory. The binary maps provided here are just an example of the possible outcome that can be obtained. A more appropriate threshold can be set as needed, resulting in a new binary risk map that accounts for the available resources and evaluated hazards.

However, it is important to take note that the carcass detection/surveillance activity is likely to have been influenced by previous ASF findings in wild boars, which may have targeted the subsequent search by veterinary/forestry services in similar areas [92], overestimating the classification power of the connectivity map. This poses an unavoidable bias inherent in the data source. On the other hand, the different nature of occurrence input data (based on availability, convenience data) compared to ASFV genotype II positive wild boar coordinates, which instead come from targeted research as described by the ASF control plans [80], may have limited binary maps classification performance. In future studies, if the epidemic front progresses, and more locations of ASF events become available, positive wild boar records could be used as presence data input, while the suitability and/or connectivity maps could be included as predictor variables, to model "deathbeds" of ASFV affected wild boars [92]. Additionally, the integration of the suitability and connectivity maps with other factors driving the disease

spreading as model variables, would allow the prediction of the epidemic directionality [120], both within the wild population and from wild boar to domestic pig.

## Implication for ASF management and surveillance strategies

Despite the aforementioned limitations, mainly related to the nature of input data, both the suitability map and connectivity map developed in this study could serve as supporting tools to be integrated into current ASF epidemic management strategies. Briefly, based on the approach of regionalisation and according to the Commission Implementing Regulation (EU) 2021/605, when an ASF case is confirmed in a wild boar in the territory, an infected zone (later defined as restriction zone II) and a surrounding surveillance zone (restriction zone I) are defined, following administrative boundaries. In these areas, active search for wild boar carcasses is performed, also with the help of dogs and drones, ideally removing infected carcasses every two weeks [80]. In fact, it is estimated that transmission between wild boar and infected carcass does not occur until 12 to 15 days [121, 122]. However, this pace is hardly maintained, except at the onset of the epidemic when its expansion is still limited.

Currently, there are no defined rules in institutional reports/regulation on where to search for carcasses to increase detection rates. Additionally, people available to actively look for carcasses are often insufficient, resulting in the search being concentrated in sloping areas where high wild boar densities are estimated, and in declivous areas [80]. However, to the best of authors' knowledge, an up-to-date, local and accurate estimate of wild boar density is often not available [37]. Studies performed in other countries suggest young, broad, moist, and cool forests, or grasslands with significant vegetation as preferred habitats for diseased wild boars, probably due to high fever, or because perceived as safer environments [92, 123]. Similar studies should be performed also in Italy, to assess if moribound wild boars behaviour is the same observed in other countries or if it is determined by local ecological drivers. The evidence on local deathbed preferences could be integrated with the risk maps obtained in this study to improve early detection of wild boar carcasses. Together with the active search for carcasses, physical barriers (natural or artificial) are built/reinforced to contain the progression of the epidemic front. However, this requires time-consuming bureaucracy often incompatible with the rapidity of the phenomenon [33]. Both suitability map and connectivity map could provide valuable support, to efficiently identify areas feature by a higher animal flow and optimize resources to, control wild boar population and disease spread. Indeed, the suitability score could provide an indirect indication of the likely wild boar density, when data are not available. The hunting activity should be focused on highest suitable area, to preventively reduce wild boar population in areas not yet infected [124–126]. However, culling strategies should be properly designed, since hunting pressure could favour wild boar migrations, increasing the risk of wild boar dispersal in ASF affected areas, and therefore ASF spreading [127, 128].

In tandem, the connectivity map would recommend on which zones/paths to target and prioritize carcass search and barrier construction of wild boar dispersal corridors. Primary disruption of corridors with higher probability of passage (higher connectivity value) through construction and reinforcement of man-made (e.g., fences, highways) or natural (e.g., rivers) barriers would efficiently contain the disease spread. On the other hand, the impact on other species should not be ignored, as landscape connectivity guarantees their preservation, migration, biodiversity, and gene flow [129, 130]. Veterinary epidemiological services should work synergistically with conservation experts to find a compromise that can preserve both the ecology of local species and animal health.

## Conclusion

The maps of suitable habitats and major corridors of wild boar spreading made available in this study represent a significant step forward in understanding and controlling ASF in Northern Italy. Having estimated the suitability and connectivity of the territory from wild boar presence data, not related to disease outbreaks, make the maps obtained in this study transversally applicable in other investigations, related to other transmissible diseases (e.g., Aujeszky disease, Classical swine fever, Tuberculosis, etc.), as well as in other research fields. Similarly, the replication of the framework here adopted in other Italian regions and other countries, necessarily represents a future perspective to act in prevention, especially considering the wild boar population density, and the related food/tourism market in some regions (e.g., Tuscany). The fruitful use of open-source presence data, collected by citizen associations and naturalists, demonstrates how citizen science is a promising approach that even indirectly could improve veterinary surveillance. Additionally, the application of species distribution methods, commonly used in the field of ecology and conservation, for the development of tools aimed at enhancing control and surveillance of diseases of veterinary concern, attests the potential of multidisciplinary approaches in addressing present and future challenges.

## Supporting information

**S1 Table. Environmental variables included in the species distribution models (SDM) for each scenario, after testing for collinearity and multicollinearity.** Variables taking into account distance from environmental features are shortened with "Dist. from", while those considering topological patterns with "Top.". Variable references are available in S2 Table. (PDF)

**S2 Table. Metadata associated to the variables included in the study.** (XLSX)

**S3 Table. Performance of algorithms of species distribution modelling (SDM), available in the package *Biomod2*, evaluated through Area Under ROC (Relative Operating Characteristic) Curve (AUC) and True Skills Statistics (TSS) metrics.** Mean and standard deviation (sd) are reported for each considered scenario. (XLSX)

**S1 Appendix. Response curves of the computed Gradient Boosted Machine ensemble model.** (PDF)

**S1 Fig. Distribution of wild boar records and estimated suitability maps by season.** (A) Distribution of occurrence records included in the final model (red points) by season. (B) Maps describing habitat suitability for the presence of wild boars in Northern Italy referring to environmental conditions by season. Base map of Italy based on the 2024 regional administrative boundaries from the Italian National Institute of Statistics (https://www.istat.it/). (TIF)

**S2 Fig. Map of the study area reporting geographical details.** Base map made with Natural Earth. Study area and Asti province based on the 2024 administrative boundaries from the Italian National Institute of Statistics (https://www.istat.it/). Lake Maggiore, Po River, regional parks details made from vector files available on the National geoportal of the Italian Ministry of Environment and Energy Security (https://gn.mase.gov.it/) for illustrative purpose only. (TIF)

**S1 File. Main code scripts used in the workflow are available at https://doi.org/10.5281/zenodo.14639171.**
(PDF)

**S2 File. Suitability maps, connectivity map and binary risk maps are available at https://doi.org/10.5281/zenodo.14637229.**
(PDF)

## Acknowledgments

Special thanks to colleague Martina Ossola (freelance veterinary doctor) for providing epidemiological data on wild boar records from the official routine culling campaigns conducted in the Varese province, in Lombardy, and to the following colleagues for their advice and help during the course of the project: Maxime Lenormand, Jérémy Lavarenne, Renaud Marti (all affiliated with TETIS Unit, INRAE, National Research Institute on Agriculture, Food and the Environment, Montpellier, France), Anas Zakroum (French Agricultural Research Centre for International Development, CIRAD, Montpellier, France), Daria Di Sabatino and Lara Savini (both affiliated with Istituto Zooprofilattico Sperimentale dell'Abruzzo e del Molise 'G. Caporale' (IZS-Teramo), Teramo, Italy), Isadora Benvegnù (Veneto Agricoltura), Mario Chiari (Direzione Generale Welfare di Regione Lombardia, Unità Organizzativa Veterinaria, Milan, Italy), Francesca Meriggi (Ente Regionale per i Servizi all'Agricoltura e alle Foreste, Milan, Italy).

## Author Contributions

**Conceptualization:** Giulia Faustini, Jaime Bosch, Annamaria Conte, Annelise Tran.

**Data curation:** Giulia Faustini, Jaime Bosch, Annelise Tran.

**Formal analysis:** Giulia Faustini, Marie Soret, Alexandre Defossez, Jaime Bosch, Annelise Tran.

**Funding acquisition:** Giulia Faustini.

**Investigation:** Giulia Faustini, Annelise Tran.

**Methodology:** Giulia Faustini, Marie Soret, Alexandre Defossez, Jaime Bosch, Annelise Tran.

**Project administration:** Giulia Faustini, Annelise Tran.

**Resources:** Giulia Faustini, Annelise Tran.

**Software:** Giulia Faustini, Marie Soret, Alexandre Defossez, Jaime Bosch, Annelise Tran.

**Supervision:** Jaime Bosch, Annamaria Conte, Annelise Tran.

**Validation:** Giulia Faustini, Jaime Bosch, Annelise Tran.

**Visualization:** Giulia Faustini.

**Writing – original draft:** Giulia Faustini.

**Writing – review & editing:** Giulia Faustini, Marie Soret, Alexandre Defossez, Jaime Bosch, Annamaria Conte, Annelise Tran.

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
