## [Decision Letter · Decision Letter 0]

17 Sep 2024

PONE-D-24-33277Habitat suitability mapping and landscape connectivity analysis to predict African swine fever spread in wild boar population: a focus on Northern ItalyPLOS ONE

Dear Dr. Faustini,

Thank you for submitting your manuscript to PLOS ONE. After careful consideration, we feel that it has merit but does not fully meet PLOS ONE’s publication criteria as it currently stands. Therefore, we invite you to submit a revised version of the manuscript that addresses the points raised during the review process.

**Dear Authors,** **the topic of the manuscript is interesting, and you are working with a valuable dataset. However, as highlighted by the reviewers, all sections require improvements. While ensemble modeling can be a powerful tool, it may also lead to misinterpretations if not applied properly. Simply adding more statistics does not necessarily result in better outcomes. Morevoer, since the main objective of your manuscript focuses on connectivity and the role of corridors, these aspects should be discussed in greater detail. I encourage you to carefully address the reviewers' constructive comments.**

We look forward to receiving your revised manuscript.

Kind regards,

Francesco Bisi, Ph.D.

Academic Editor

PLOS ONE

**Journal Requirements:**

The research project was made possible by the financial support provided to G.F. by the Società Italiana di Patologia ed Allevamento dei Suini (SIPAS, https://www.sipas.org/chi-siamo/) and by the Erasmus + for traineeship mobility program (https://erasmus-plus.ec.europa.eu/opportunities/opportunities-for-individuals/students/traineeships-abroad-for-students). The funders had no role in study design, data collection and analysis, decision to publish, or preparation of the manuscript.

3. Please note that your Data Availability Statement is currently missing the repository name and the DOI/accession number of each dataset OR a direct link to access each database. If your manuscript is accepted for publication, you will be asked to provide these details on a very short timeline. We therefore suggest that you provide this information now, though we will not hold up the peer review process if you are unable.

Reviewers' comments:

Reviewer's Responses to Questions

**Comments to the Author**

1. Is the manuscript technically sound, and do the data support the conclusions?

Reviewer #1: Partly

Reviewer #2: Yes

2. Has the statistical analysis been performed appropriately and rigorously? 

Reviewer #1: Yes

Reviewer #2: Yes

3. Have the authors made all data underlying the findings in their manuscript fully available?

Reviewer #1: Yes

Reviewer #2: Yes

4. Is the manuscript presented in an intelligible fashion and written in standard English?

Reviewer #1: No

Reviewer #2: Yes

5. Review Comments to the Author

**Reviewer #1:** I appreciate the effort of the authors to pull together this information, however, evaluating all possible models in the Biomod2 package does not seem like an ideal study design. Even more confusing is the reason for choosing GBM as more preferred. The authors have a suitable dataset to use a few models for their purposes. With their knowledge and data available, a model or even a few could be chosen but certainly 11 models are not needed to be evaluated. Then we have to delve into the evaluation criteria for why the other 8 models are not suitable which just detracts from the focus of your study design..predicting ASF across the landscape.

In my comments below, I tried to assist the authors on how to address some of these issue so their objectives and methods are more clear and support their findings with some specific comments by line:

Title: Habitat suitability mapping and landscape connectivity analysis to predict African swine fever spread in wild boar population: a focus on northern Italy. I believe populations should be plural or place an “a” before “wild boar” if authors are suggesting this is one single wild population throughout their study site where records are documented?

Introduction

Lines 44 and 51: Defining African swine fever as ASF and African swine fever virus as ASFV seems confusing and unnecessary. The authors then go into ASFV genotype I and ASFV genotype II but reference ASF to discuss the disease in general terms. It is difficult for the reader to know which is meant or which is most appropriate each time either are used. I would suggest do not define ASFV and refer to these as genotype I and genotype II each time it is needed. Then use ASF when discussing the disease in general. Furthermore, there is no mention of these in the remainder of the manuscript aside from ASF.

Lines Line 116: “different wild boar presence data types” is very confusing as written. I would suggest “Presence/absence data for wild boar differs considerably, although often in relation to a common/similar set of environmental variables describing topography, climate, human disturbance and land cover [29].” The authors then need to cite more than one manuscript if these types do differ in the literature.

Line 122: ENET needs to be spelled out prior to using an acronym because most won’t be familiar with this term.

Line 128: “..dispose of..” I am not sure what this means? This entire sentence is very confusing and might be better split into 2 separate sentences.

Materials and methods

Line 177: Supplement Table S1. This information is often overlooked and I appreciate this table. This is the most complete and detailed variable table I have ever reviewed, thank you for this great summary table!

Lines 170-188: A few points of clarity would be helpful in this section.

1. Density – are these within a 100 x 100 m raster that lines up with all of your rasters? If not, how do you get percentage assigned to each raster cell for bare, herbaceous, and tree cover?

2. Bilinear sampling of 30x30 m raster to derive 100x100m rasters? If so, this should be presented in your “percentage of forest” in your S1 Table.

3. By “overall mean” do you mean “annual mean” for each year from 2014-2023 or a single static 10 year mean from all years: 2014-2023?

a. A mean for a 10 year window, if that is what was done, is of no value so please be clear here. Each year of wild boar data should be matched to variables when available. Specifically, NDVI could have massive fluctuations from year to year so a 10 year average does not seem acceptable in any way?

4. What are the “seasons” based on? Some citations for seasons in northern Italy would be advisable.

Lines 195-197: What variables were removed due to collinearity and multicollinearity? The authors state in line 213 that “Only non-collinear variables were incorporated” but these should be listed here in the methods if they were not included. Then delete Line 213? I see now these are in Table 1 (better to reference in Methods) but this table can be deleted. Simply include a sentence of which are not included because they don’t meet assumptions of collinearity/multicollinearity (i.e., opposite of Lines 328-332) in the methods not results.

Lines 227-232: In your introduction, the authors discuss that SDMs can be implemented based on data available and aim of the study. However, here the authors appear to go on a habitat suitability modeling exploration with everything in Biomod2 (11 methods) which is inappropriate and unnecessary post hoc analysis that goes against the authors own introduction. Why not select the best model that fits your data and use it instead of evaluating which is best based on some criteria that we really are not clear on? More on this later.

Results

Line 341: Here the authors refer to “annual” not “overall” leading to confusion on just how these were calculated?

Line 347: I am a bit confused here. In the Introduction, the authors were recommending an “ensemble modelling approach” but here they are saying just use one, GBM? So this entire effort of evaluating all of these models was not necessary?

Line 348: See my comments above for Lines 227-232 and Line 347. The authors claim GBM was preferred due to “easier parameterization” and “more realistic performance metrics?” I am not sure how you define this or justify it being a criteria for selecting the model chosen?

Line 347: What is a “privileged area?”

Figure 1-4: These figures appear blurry and may not be up to the quality required by the journal.

Figure 3: It seems to me that the suitability of habitat lines up with where the presence data is located for this region, regardless of season?

Figure 4: It is difficult to determine this figure for panel A. I suggest you zoom in on the connectivity map like in B and C but replace those panels with connectivity values instead of binary risk?

Discussion

Lines 4342-436: I am not sure the authors succeeded here based on my point above that the suitability and connectivity seem to follow the presence data? Using a 100 m resolution also seems too fine a scale considering the size of their study area. There may be too similar characteristics across the study area at the 100 m scale thus masking the variables that were really influencing wild boar distribution across your landscape? A larger scale, with your variables summarized to each raster cell (500m or 1000 m) might provide more value and identify more influential variables in your models?

Lines 442-449: Agreed, does the lack of low suitability in Po Valley make sense for how wild boar select habitat in Europe? Are there any detailed VHF or GPS data collected in this region to determine if wild boar prefer agriculture-forest valleys compared to forested, mountainous terrain?

Lines 484-486: The authors acknowledge the temporal pattern but it was not clear how these were modeled with “time” as stated in the methods? All climate variables are available daily, weekly, monthly correct? However, the authors used a static annual mean or 3-4 months seasonal mean which likely rendered these variables of limited value in their assessment. These climate variables and NDVI could have avoided these pitfalls the author outline if they were evaluated differently here?

**Reviewer #2: **Faustini et al. presented a ms entitled “Habitat suitability mapping and landscape connectivity analysis to predict African swine fever spread in wild boar population: a focus on Northern Italy” for publication in PLOS ONE. The paper deals with a relevant and interesting topic: a highly contagious disease affecting wild and domestic pigs (ASF) reported in Eastern Europe and, in January 2022, in North-western Italy. Thereafter, the expanding wild epidemic, has posed serious management problems related to the presence of Italian pig farms and trade.

I enjoyed reading this ms and I think that this study is a valuable contribution to improve the knowledge/management of this species.

However, despite the interesting topic and the potential of management implications for the target species (i.e., mapping of suitable habitats for wild boars and their potential dispersal corridors in Northern Italy, could be an important issue in preventing the ASF expansion), I believe that a revision is required (please, see my comments below).

General comments:

1) A first flaw that I found concerns the absence of an accurate description of ASF Genotype II expansion (speed, population crash, corridors) in the Eastern European countries. I believe this is a fundamental point: to better introduce the issue, the authors should describe the situation of the last 10 years in Eastern Europe in detail, also in relation to the economic impact of this epidemic.

2) I think the ms would gain more value with a reorganization of the discussion section. I suggest dealing first with the description of the choices made by wild boar, then the description of the potential expansion corridors (with details on the most important directions and their characteristics) and finally an analysis of the possible management and surveillance interventions. These three parts are already present in the discussion but should be better outlined, especially the part related to connectivity.

3) A detailed explanation/comparison of seasonal models is also missing. How does expansion relate to seasons?

4) The manuscript often reports the names of valleys, rivers, mountain areas with reference to the figures (e.g., Fig. 3 - L. 442-446) but these are not reflected in the maps. For a reader who is not familiar with these areas, it is difficult to interpret the results. Supplementary material with more detailed maps could be added.

5) Also the “discontinuous natural zones” (L. 471) should be represented in a map, according to their importance in connectivity

Minor comments:

Abstract: I suggest to delete the description of ASF Genotype I that is not the target of this ms.

L.439-441: “In addition to the higher resolution of the maps presented here compared to those found in the literature, the influence of each variable on wild boar presence was also detailed.”

Please include the references of the studies compared with the results obtained in this research.

L. 507-510: I think that high-traffic corridors need to be described: characteristics, directions and relationships with existing farms.

6. PLOS authors have the option to publish the peer review history of their article (what does this mean?). If published, this will include your full peer review and any attached files.

Reviewer #1: No

Reviewer #2: No

---

## [Author Response · Author response to Decision Letter 0]

5 Nov 2024

Response to Academic Editor 

Dear Authors,

the topic of the manuscript is interesting, and you are working with a valuable dataset. However, as highlighted by the reviewers, all sections require improvements. While ensemble modeling can be a powerful tool, it may also lead to misinterpretations if not applied properly. Simply adding more statistics does not necessarily result in better outcomes. Morevoer, since the main objective of your manuscript focuses on connectivity and the role of corridors, these aspects should be discussed in greater detail. I encourage you to carefully address the reviewers' constructive comments.

A: We thank the academic editor for the positive feedback and for highlighting key areas for improvement. We have addressed each of the reviewers' comments and incorporated their suggestions into the revised manuscript. Specifically, we have better clarified our approach, clarifying its application and limitations to avoid any potential misinterpretations. Additionally, as recommended, we have enriched the discussion of connectivity and corridor roles, detailing how these features impact wild boar movement and potential disease spread across the landscape. 

Moreover, the following adjustments were made in figures, tables based on the reviewers' valuable suggestions:

• Figure 3: We now present only the overall habitat suitability map, with seasonal results provided separately as S1 Fig.

• Figure 4: As suggested by Reviewer 1, we included a zoomed-in view of infected areas in the connectivity map to enhance focus on regions most affected by African swine fever.

• S2 Fig: We added a new figure illustrating detailed geographical context, addressing Reviewer 2’s request.

• Table 1: We clarified the methodology by adding a description of the standardization process for the raster data, as suggested by Reviewer 1.

• Data Availability: Following the suggestions for data transparency, we have made all associated scripts (S1 Files) and original raster files (S2 Files) available via GitLab and Google Drive links, with the files set to be publicly accessible upon acceptance of the manuscript.

• Corresponding Author Contact: We included an additional email for the corresponding author to accommodate a potential upcoming transition to a new institutional affiliation at the University of Padova.

• Funding Statement: the funding acknowledgment now reads: "The research project was made possible by the financial support provided to G.F. by the Società Italiana di Patologia ed Allevamento dei Suini (SIPAS, https://www.sipas.org/chi-siamo/) and by the Erasmus + for traineeship mobility program (https://erasmus-plus.ec.europa.eu/opportunities/opportunities-for-individuals/students/traineeships-abroad-for-students). The funders had no role in study design, data collection and analysis, decision to publish, or preparation of the manuscript. There was no additional external funding received for this study."

We hope to have addressed each of the reviewers' feedback comprehensively, and we thank them for helping us in refining the manuscript. We look forward to your further feedback.

Response to Reviewers

Reviewer #1: I appreciate the effort of the authors to pull together this information, however, evaluating all possible models in the Biomod2 package does not seem like an ideal study design. Even more confusing is the reason for choosing GBM as more preferred. The authors have a suitable dataset to use a few models for their purposes. With their knowledge and data available, a model or even a few could be chosen but certainly 11 models are not needed to be evaluated. Then we have to delve into the evaluation criteria for why the other 8 models are not suitable which just detracts from the focus of your study design..predicting ASF across the landscape.

A: We thank the reviewer for the valuable comments, and hope that our revisions will address his/her concerns effectively. Please see below.

In my comments below, I tried to assist the authors on how to address some of these issue so their objectives and methods are more clear and support their findings with some specific comments by line:

A: We would like to sincerely thank the reviewer for the insightful and constructive comments. We truly appreciate the time and effort taken to provide suggestions on our manuscript. 

Title: Habitat suitability mapping and landscape connectivity analysis to predict African swine fever spread in wild boar population: a focus on northern Italy. I believe populations should be plural or place an “a” before “wild boar” if authors are suggesting this is one single wild population throughout their study site where records are documented?

A: We thank the reviewer for the suggestion. We modified the title accordingly, to improve it clarity. 

Introduction

Lines 44 and 51: Defining African swine fever as ASF and African swine fever virus as ASFV seems confusing and unnecessary. The authors then go into ASFV genotype I and ASFV genotype II but reference ASF to discuss the disease in general terms. It is difficult for the reader to know which is meant or which is most appropriate each time either are used. I would suggest do not define ASFV and refer to these as genotype I and genotype II each time it is needed. Then use ASF when discussing the disease in general. Furthermore, there is no mention of these in the remainder of the manuscript aside from ASF.

A: We thank the reviewer for the valuable suggestion to improve the clarity of our manuscript. Following reviewer’s suggestion, the introduction was structured presenting at the beginning only the disease as caused by the virus, and only later the virus with its genotypes. We preferred to keep the distinction between genotype I and II, to present the different epidemiological scenarios of Sardinia Island compared to the rest of Europe and Italy (mainland). However, we acknowledge reviewer observation that ASFV genotypes I and II are not referenced in the remainder of the manuscript. In the revised version, we ensured that any reference to ASF cases specifies the genotype II, including a general statement in the “model evaluation” section in line 333-334. 

Lines Line 116: “different wild boar presence data types” is very confusing as written. I would suggest “Presence/absence data for wild boar differs considerably, although often in relation to a common/similar set of environmental variables describing topography, climate, human disturbance and land cover [29].” The authors then need to cite more than one manuscript if these types do differ in the literature.

A: We thank the reviewer for the suggestion. We modified the manuscript accordingly, citing also more articles. 

Line 122: ENET needs to be spelled out prior to using an acronym because most won’t be familiar with this term.

A: We acknowledge that this term may appear to be an acronym, but it is, in fact, the official name of the project/consortium. Unfortunately, there is no acronym definition provided by the European Food Safety Authority (EFSA). Instead, EFSA defines ENETWILD as an international network of wildlife professionals, which we had described in line 142-143 as "an international network of wildlife professionals supported by the European Food Safety Agency." We apologize for any confusion and hope this clarification resolves the issue.

Line 128: “..dispose of..” I am not sure what this means? This entire sentence is very confusing and might be better split into 2 separate sentences.

A: We apologize for the misleading sentence. The sentence has been revised to enhance its clarity. Specifically, we have replaced "dispose of" with "would benefit from" and have split the sentence into two parts for better readability (see lines 148-151). 

Materials and methods

Line 177: Supplement Table S1. This information is often overlooked and I appreciate this table. This is the most complete and detailed variable table I have ever reviewed, thank you for this great summary table!

A: We sincerely thank the reviewer for appreciating our work and effort! We wanted to provide something helpful for improving the reproducibility of our methods.

Lines 170-188: A few points of clarity would be helpful in this section.

1. Density – are these within a 100 x 100 m raster that lines up with all of your rasters? If not, how do you get percentage assigned to each raster cell for bare, herbaceous, and tree cover?

A: Yes, all the rasters were standardized to 100x100m grid cells using a bilinear sampling method , including the coverage layers, as explained in lines 200-203. To avoid confusion, we add this specification also in the description of each variable in S1 Table. 

2. Bilinear sampling of 30x30 m raster to derive 100x100m rasters? If so, this should be presented in your “percentage of forest” in your S1 Table.

A: All rasters were standardized to 100x100m grid cells, as explained in lines 200-203 in material and methods section. We are not sure what the reviewer refers to as there aren’t any variables that were standardized from 30x30 m raster to derive 100x100m raster. However, to avoid confusion we add the specification “Bilinear sampling method was applied to standardize the variable as raster with a pixel size of 100 m x 100 m, over an extent equal to the study area.” also in the description of each variable in S1 Table. We thank the reviewer for the opportunity to clarify better our methodology to the reader. 

3. By “overall mean” do you mean “annual mean” for each year from 2014-2023 or a single static 10 year mean from all years: 2014-2023?

a. A mean for a 10 year window, if that is what was done, is of no value so please be clear here. Each year of wild boar data should be matched to variables when available. Specifically, NDVI could have massive fluctuations from year to year so a 10 year average does not seem acceptable in any way?

A: We thank the reviewer for highlighting this important point. To clarify, when we refer to "overall mean," we are specifically addressing environmental/climatic variables, which represent long-term environmental patterns rather than short-term weather fluctuations. In niche modeling, the objective is to capture stable climatic conditions that shape habitat suitability over extended periods, rather than ephemeral weather-driven variations. Generally, climatic data spanning even 30 years is standard practice for such analyses. 

Therefore, we respectfully disagree with the suggestion that a 10-year mean holds no value in this context. We believe that our approach is well-suited to the scale and aims of our study, focused more on the general suitability of the landscape for wild boars (which is a highly adaptable species), rather than the year-to-year variability that might arise from short-term shifts. 

Our approach, which involves calculating an overall mean for the 2014–2023 period, was chosen deliberately to smooth out short-term fluctuations and provide a more stable representation of the environmental/climatic conditions affecting wild boar populations over time. While annual data may indeed capture year-to-year variability, such as NDVI fluctuations, our primary focus was on long-term trends and consistent drivers of wild boar presence. Using a 10-year mean allowed us to generalize the broader environmental context, providing a more robust measure of habitat conditions that is less sensitive to short-term fluctuations. Furthermore, it is not uncommon in ecological and environmental studies to use multi-year averages for environmental/climatic variables, particularly considering data availability and when the goal is to account for general patterns over an extended time frame rather than year-to-year variability (here some examples: https://doi.org/10.1371/journal.pone.0193295, https://doi.org/10.1371/journal.pone.0176339, https://doi.org/10.1007/s10980-021-01371-y). 

That said, we acknowledge the limitations of our method, particularly as pointed out by the reviewer and that using year-specific NDVI could enhance temporal precision in future studies focused on annual fluctuations and dynamics. We added this points in the manuscript in line 208-210 and 514-522. 

4. What are the “seasons” based on? Some citations for seasons in northern Italy would be advisable.

A: The seasons were defined based on the meteorological convention, which divides the year into four three-month periods: December, January, and February for winter; March, April, and May for spring; June, July, and August for summer; and September, October, and November for autumn. This method does not account for shorter units of time, such as weeks or days, and was used for its simplicity and convenience. Unlike the astronomical definition, which aligns the seasons with solstices and equinoxes, the meteorological definition directly reflects temperature and weather patterns, which are more relevant for understanding seasonal dynamics in wild boar distribution. We clarified this choice in the manuscript and provide references to support the use of this definition (line 211-213). 

Lines 195-197: What variables were removed due to collinearity and multicollinearity? The authors state in line 213 that “Only non-collinear variables were incorporated” but these should be listed here in the methods if they were not included. Then delete Line 213? I see now these are in Table 1 (better to reference in Methods) but this table can be deleted. Simply include a sentence of which are not included because they don’t meet assumptions of collinearity/multicollinearity (i.e., opposite of Lines 328-332) in the methods not results.

A: We thank the reviewer for the helpful feedback. We have moved the description of the included variables, as well as the corresponding Table 1, into the methods section to improve clarity (lines 224-230). We acknowledge that Table 1 could seem a repetition with the text. However, we decided to maintain Table 1 as we think it provides a clear and immediate overview of the variables, making it easier for readers to reference at a glance. Keeping the table also allows us to maintain a logical and seamless flow in the result subsection "presence data and environmental variables" referring to the methods, ensuring the text remains easy to follow. We appreciate reviewer’s input in helping to enhance the clarity and linearity of the manuscript.

Lines 227-232: In your introduction, the authors discuss that SDMs can be implemented based on data available and aim of the study. However, here the authors appear to go on a habitat suitability modeling exploration with everything in Biomod2 (11 methods) which is inappropriate and unnecessary post hoc analysis that goes against the authors own introduction. Why not select the best model that fits your data and use it instead of evaluating which is best based on some criteria that we really are not clear on? More on this later.

A: We apologize for the confusion and thank the reviewer for the valuable comment. About evaluating all possible models in the Biomod2 package, we completely understand that this may seem excessive. However, since behind the choice of a model there are many considerations and assumptions, choosing the best performing model by comparing multiple ones is an accepted approach when SDMs are considered (https://doi.org/10.1002/ece3.6859; https://doi.org/10.1111/1749-4877.12000; https://doi.org/10.1007/s10980-021-01371-y; https://doi.org/10.1016/j.ecoleng.2022.106725). This rationale was suggested also by ENETWILD report (https://doi.org/10.2903/sp.efsa.2018.EN-1490), at page 12: “In order to provide a best possible output we would argue that all potential models should be fitted and compared to limit the risk of artefactual outputs as a result of model-specific assumptions”. Comparing a variety of models allowed us to identify the best-performing one for our dataset. We wanted to avoid making assumptions about which models would be most suitable, and instead, allow the data to guide the selection process. We added a specification in the discussion section in line 560-561 to support our rationale.

However, we agree on the confusing introduction. Thus, we modified it accordingly, also with some references, in line 140-145.

Results

Line 341: Here

---

## [Decision Letter · Decision Letter 1]

4 Dec 2024

PONE-D-24-33277R1Habitat suitability mapping and landscape connectivity analysis to predict African swine fever spread in wild boar population: a focus on Northern ItalyPLOS ONE

Dear Dr. Faustini,

Thank you for submitting your manuscript to PLOS ONE. After careful consideration, we feel that it has merit but does not fully meet PLOS ONE’s publication criteria as it currently stands. Therefore, we invite you to submit a revised version of the manuscript that addresses the points raised during the review process.

ACADEMIC EDITOR: the manuscript requires minor revisions. Please carefully follow my comments below. 

We look forward to receiving your revised manuscript.

Kind regards,

Francesco Bisi, Ph.D.

Academic Editor

PLOS ONE

Journal Requirements:

**Additional Editor Comments:**

Dear Authors,

I would like to thank you for considering and addressing most of the reviewers' comments. I believe the manuscript contributes significantly to the understanding of a very interesting and sensitive subject.

I must say that given the wide range of potential applications for suitability models, there may never be a "perfect" model or an entirely "wrong" one. Furthermore, due to the complexities of wild boar ecology, modeling its distribution will always present challenges. This species in Europe can be found in very different habitats, ranging from urban areas to mountainous regions up to 2000 m a.s.l. Initially, as Reviewer 1, I was skeptical about the use of NDVI data averaged over a ten-year period. However, I now realize that investigating the subject in greater detail would involve a vast array of additional covariates (e.g., beech and chestnut masting, snow cover, predator densities, hunting pressure), making the endeavor endless. Thus, I consider your selection of covariates appropriate.

However I would like to request a few additional improvements:

As suggested by Reviewer 1, please carefully confirm the number of variables used in the modeling process 11 or 13? Additionally, I agree with Reviewer 1 that Table 1 does not add substantial value to the main text and should be moved to the supplementary material.

Please carefully review the references, as suggested by Reviewer 1. For example, while working with grids in R, I currently use the "Terra" package, but I understand that the "Raster" package is still functional for older versions of R. Kindly verify which package was utilized and ensure the references reflect this accurately.

While reading the manuscript, I felt that the seasonal models were not well integrated into the overall narrative. It would be helpful to clarify why you chose to model seasonal distributions and to better explain why the finding of increased suitability in spring could be of any interest. Does this have any management implications?

Reviewers' comments:

Reviewer's Responses to Questions

**Comments to the Author**

1. If the authors have adequately addressed your comments raised in a previous round of review and you feel that this manuscript is now acceptable for publication, you may indicate that here to bypass the “Comments to the Author” section, enter your conflict of interest statement in the “Confidential to Editor” section, and submit your "Accept" recommendation.

Reviewer #1: (No Response)

Reviewer #2: All comments have been addressed

2. Is the manuscript technically sound, and do the data support the conclusions?

Reviewer #1: Partly

Reviewer #2: Yes

3. Has the statistical analysis been performed appropriately and rigorously? 

Reviewer #1: No

Reviewer #2: Yes

4. Have the authors made all data underlying the findings in their manuscript fully available?

Reviewer #1: No

Reviewer #2: Yes

5. Is the manuscript presented in an intelligible fashion and written in standard English?

Reviewer #1: Yes

Reviewer #2: Yes

6. Review Comments to the Author

Reviewer #1: I appreciate the effort of the authors to pull together this information, the manuscript has been improved considerably in the Introduction specifically, and I appreciate the authors efforts in addressing reviewer concerns. There were still a few concerns that were not addressed and one that I overlooked that I will go into detail below:

Materials and methods

Lines 215-217: What is reference 67 actually referring to here? There is no mention of a correlation tree or cluster dendrogram or raster package in the document? How can the raster package have been used when it was removed from CRAN for use in program R in December 2023? It appears the authors are citing a review (Anderson; Journal of Experimental Marine Biology and Ecology 289(2):303–305) of the book:

Analysis of Ecological Communities: Bruce McCune and James B. Grace, MjM Software Design, Gleneden Beach, USA, 2002, ISBN 0 9721290 0 6, US$ 35 (Pbk) May 2003.

Regardless, none of these terms appear in the book either as far as I can tell.

Line 224: “The variables included in the study for all scenarios were 11:…” Do the authors mean “overall” here instead of “all” to align with text and table 1 that refers to all seasons combined (Line 207)?

Lines 224-228: In my previous review, I suggested “Simply include a sentence of which are not included because they don’t meet assumptions of collinearity/multicollinearity (i.e., opposite of Lines 328-332) in the methods not results.” The authors chose to not edit with my recommendations and also chose not to delete Table 1. Perhaps this time around I might provide a more compelling argument to deleting (or modifying) Table 1 due to the number of issues with it’s inclusion:

1. Out of 20 variables, 11 were acceptable to include in the overall analysis although Table 1 has x in 13 columns along with the total that confirms the 13 and not the 11?

2. The authors list the 11 here as well as several that were not included in overall or seasonal models which, in turn, is presenting "results" in the Materials and Methods? There is also a section in the Results (Lines 365-368) that says out of the 20 variables that 13 were selected in the overall and seasonal suitability models that references Table 1. It is not clear why some of this analysis is in the Materials and Methods and some is in Results?

3. Lines 226-228 identifies several variables that were excluded in all scenarios (overall and seasonal, I presume), and interestingly, 2 of the variables (Temperature and NDVI) are those averaged over a 10 year span that I mentioned was not appropriate in my previous review (see below comments from my previous review and author response).

4. Table 1 repeats some of what is in text in both sections and I would still suggest that it be deleted. If the authors believe it has value, then why not include mean (SDs) for each variable during each season and overall instead of just an x? An entire page for a table with x’s that could be included in 4 sentences in the Materials and Methods (like Lines 226-228 so one line for overall and one line for each season) does not seem necessary and I don’t see how it helps the “flow” considering it is also presented in the Results?

From my previous review:

Lines 484-486 (original submission): The authors acknowledge the temporal pattern but it was not clear how these were modeled with “time” as stated in the methods? All climate variables are available daily, weekly, monthly correct? However, the authors used a static annual mean or 3-4 months seasonal mean which likely rendered these variables of limited value in their assessment. These climate variables and NDVI could have been match with each week-month-year presence data were recorded?

To which the authors responded:

A: While it is true that climate variables are available at more granular temporal resolutions (daily, weekly, and monthly), our decision to employ annual or seasonal means was driven by the objective of focusing on broader climate trends relevant to wild boar habitat suitability. We added a sentence in the discussion in line 556-559 to recognize that a more dynamic approach, utilizing daily or weekly variables, could provide a richer understanding of how seasonal variability may influence habitat selection over time. We thank the reviewer for highlighting this important aspect, as it encourages us to refine our methodologies and consider more nuanced evaluations in subsequent studies.

Reviewer second response: Again, I suggested the authors assign temperature and NDVI to the monthly or even seasonal means instead of a 10 year mean (overall or seasonal). The authors identified, in their response and in the manuscript that a “more dynamic approach, utilizing daily or weekly variables, could provide a richer understanding…” however the authors chose not to do it in their revision? Instead, they argue how it can be done in the future? Furthermore, do the authors realize that both Temperature and NDVI were not included in any models (Based on Table 1 multicollinearity) and Precipitation was include in 3 of the 5 scenarios only? Regardless of what past research claims, the authors have the ability to match these variables to the annual/seasonal means to match them to the data but chose not to do so. The authors might be better off removing them entirely in the manuscript and models considering they provide no value based on how they were presented in the models (perhaps due to static 10 year averages)? Based on how you present them in Table 1, Temp and NDVI were not included in any models, correct?

Supporting information, S1 Files: This Gitlab repository does not exist or is not accessible to the reviewer. The authors agreed that it should be available “upon acceptance,” however, not available to the reviewer? It seems that accepted or not, a GitLab repo would be valuable to document your work and provide this analysis as an archive for your efforts. Including this link in your revisions has little value if a reviewer is not able to access it?

Reviewer #2: This revised version has improved a lot and accounted for most of the problems I identified when reading the first version.

I think the authors have made a considerable effort to increase the quality of their ms.

7. PLOS authors have the option to publish the peer review history of their article (what does this mean?). If published, this will include your full peer review and any attached files.

Reviewer #1: No

Reviewer #2: No

---

## [Author Response · Author response to Decision Letter 1]

30 Dec 2024

Response to Editor and Reviewers

Journal Requirements:

A: We thank for the reminder to review the reference list. We confirm that the manuscript does not contain any retracted papers. Based on Reviewer 1's comments, we have updated reference 67 (now reference 60) to accurately reflect the correct source. The details of this modification are described in our response to Reviewer 1.

Additional Editor Comments:

Dear Authors,

I would like to thank you for considering and addressing most of the reviewers' comments. I believe the manuscript contributes significantly to the understanding of a very interesting and sensitive subject.

I must say that given the wide range of potential applications for suitability models, there may never be a "perfect" model or an entirely "wrong" one. Furthermore, due to the complexities of wild boar ecology, modeling its distribution will always present challenges. This species in Europe can be found in very different habitats, ranging from urban areas to mountainous regions up to 2000 m a.s.l. Initially, as Reviewer 1, I was skeptical about the use of NDVI data averaged over a ten-year period. However, I now realize that investigating the subject in greater detail would involve a vast array of additional covariates (e.g., beech and chestnut masting, snow cover, predator densities, hunting pressure), making the endeavor endless. Thus, I consider your selection of covariates appropriate.

A: We thank the Editor for the kind words and the encouraging feedback. We appreciate the recognition of the challenges inherent in modeling the distribution of such an adaptable and widely dispersed species as the wild boar. 

However I would like to request a few additional improvements:

As suggested by Reviewer 1, please carefully confirm the number of variables used in the modeling process 11 or 13?

A: Thank you for pointing this out. For each scenario, 13 variables were selected for the modeling process, 11 of which were common across all scenarios. We apologize for the misunderstanding and have revised the manuscript accordingly to make this point clearer (lines 225-230). We appreciate editor and reviewers’ attention and hope this resolves the confusion.

Additionally, I agree with Reviewer 1 that Table 1 does not add substantial value to the main text and should be moved to the supplementary material.

A: We now acknowledge that Table 1 does not add substantial value to the main text. Accordingly, we have moved Table 1 to the supplementary material as recommended (now S1 Table). We appreciate Editor and reviewer 1 input, which has helped improve the clarity and focus of the manuscript.

Please carefully review the references, as suggested by Reviewer 1. For example, while working with grids in R, I currently use the "Terra" package, but I understand that the "Raster" package is still functional for older versions of R. Kindly verify which package was utilized and ensure the references reflect this accurately.

A: We thank the editor for highlighting the importance of accurate referencing. Upon review, we realized that there was an oversight in our citation. We sincerely apologize for the confusion.

The functions we utilized for the correlation tree belong to the stats package in R, and not the Raster package as previously cited. We have corrected this in the manuscript and updated the references accordingly. Additionally, we have verified all other references to ensure their accuracy.

We greatly appreciate editor and reviewer 1 attention to detail, which has helped us improve the quality and precision of our work.

While reading the manuscript, I felt that the seasonal models were not well integrated into the overall narrative. It would be helpful to clarify why you chose to model seasonal distributions and to better explain why the finding of increased suitability in spring could be of any interest. Does this have any management implications?

A: We thank the editor for the observation and have revised the manuscript to better explain the rationale for modeling seasonal distributions.

We chose to model seasonal distributions to account for the variations in habitat use and resource availability for wild boars across seasons. We specified this choice in materials and methods section, in the lines 205-207 and lines 233-234. Specifically, the finding of increased suitability in spring suggests a period of heightened activity and resource exploitation, which is likely influenced by factors such as vegetation growth and reproductive cycles (line 494-500).

From a management perspective, these findings could inform targeted strategies, such as timing interventions to optimize hunting schedules, disease surveillance plans, and the implementation of biosecurity measures. To clarify this connection, we have added this specification to line 500-502 in the revised manuscript (discussion section).

We appreciate Editor feedback, which has allowed us to strengthen the narrative and highlight the practical implications of our findings.

Reviewers' comments:

Reviewer's Responses to Questions

Comments to the Author

1. If the authors have adequately addressed your comments raised in a previous round of review and you feel that this manuscript is now acceptable for publication, you may indicate that here to bypass the “Comments to the Author” section, enter your conflict of interest statement in the “Confidential to Editor” section, and submit your "Accept" recommendation.

Reviewer #1: (No Response)

Reviewer #2: All comments have been addressed

2. Is the manuscript technically sound, and do the data support the conclusions?

Reviewer #1: Partly

Reviewer #2: Yes

3. Has the statistical analysis been performed appropriately and rigorously?

Reviewer #1: No

Reviewer #2: Yes

4. Have the authors made all data underlying the findings in their manuscript fully available?

Reviewer #1: No

Reviewer #2: Yes

5. Is the manuscript presented in an intelligible fashion and written in standard English?

Reviewer #1: Yes

Reviewer #2: Yes

6. Review Comments to the Author

Reviewer #1: I appreciate the effort of the authors to pull together this information, the manuscript has been improved considerably in the Introduction specifically, and I appreciate the authors efforts in addressing reviewer concerns. 

A: We thank the reviewer for the kind words, and we are pleased to read that the revisions have significantly improved the manuscript. We appreciated reviewer 1's thoughtful comments and constructive suggestions, which were instrumental in enhancing the quality of the manuscript.

There were still a few concerns that were not addressed and one that I overlooked that I will go into detail below:

A: We thank the reviewer for the comments. We carefully considered the points that reviewer 1 has raised, and provided a detailed response to each one below, trying to meet each request.

Materials and methods

Lines 215-217: What is reference 67 actually referring to here? There is no mention of a correlation tree or cluster dendrogram or raster package in the document? How can the raster package have been used when it was removed from CRAN for use in program R in December 2023? It appears the authors are citing a review (Anderson; Journal of Experimental Marine Biology and Ecology 289(2):303–305) of the book:

Analysis of Ecological Communities: Bruce McCune and James B. Grace, MjM Software Design, Gleneden Beach, USA, 2002, ISBN 0 9721290 0 6, US$ 35 (Pbk) May 2003.

Regardless, none of these terms appear in the book either as far as I can tell.

A: We thank the reviewer for pointing out the oversight regarding the reference now on lines 216–218. The reviewer is absolutely correct that there was an error in our citation. We sincerely apologize for the mistake and any confusion it may have caused.

The functions used for the correlation tree actually belong to the stats package in R, and we have corrected the manuscript to reflect this (line 218). As reported by “> citation("stats")” function: “The ‘stats’ package is part of R. To cite R in publications use: R Core Team (2023). _R: A Language and Environment for Statistical Computing_. R Foundation for Statistical Computing, Vienna, Austria. https://www.R-project.org/.”

Additionally, we have included the appropriate reference for the Variance Inflation Factor (VIF) to enhance reproducibility (line 221).

We are grateful for the reviewer’s careful review and for bringing this to our attention, allowing us to improve the accuracy and clarity of our work.

Line 224: “The variables included in the study for all scenarios were 11:…” Do the authors mean “overall” here instead of “all” to align with text and table 1 that refers to all seasons combined (Line 207)?

A: We apologize for the potential confusion regarding the use of "all" versus "overall". In the previous review, we verified that the term "overall" was consistently used to avoid ambiguity. In this section, we decided instead to retain "all" because it specifically refers to the fact that for each scenario, 13 variables were selected for the modeling process, 11 of which were common across all scenarios. Table 1 (now S1 Table) served instead as a visual summary to complement this explanation. 

We apologize for any misunderstanding and have revised the manuscript to avoid further misunderstandings (line 225). 

Lines 224-228: In my previous review, I suggested “Simply include a sentence of which are not included because they don’t meet assumptions of collinearity/multicollinearity (i.e., opposite of Lines 328-332) in the methods not results.” The authors chose to not edit with my recommendations and also chose not to delete Table 1. Perhaps this time around I might provide a more compelling argument to deleting (or modifying) Table 1 due to the number of issues with it’s inclusion:

A: We thank the reviewer for the thoughtful feedback and for providing a more detailed explanation regarding his/her previous suggestion. We apologize for not fully addressing the reviewer’s earlier recommendation, and we appreciate his/her patience in this matter.

We now understand the importance of simply including a sentence in the methods section to explain which variables were excluded due to assumptions of collinearity/multicollinearity. We have revised the manuscript accordingly, incorporating this clarification as suggested in lines 225-230.

Regarding Table 1, we have reconsidered your suggestion, and, after further evaluation of editor’s comment, we have decided to move Table 1 to the supplementary material to better align with the focus of the main text.

We are grateful for reviewer constructive comments, which have been instrumental in improving the clarity and quality of our manuscript.

1. Out of 20 variables, 11 were acceptable to include in the overall analysis although Table 1 has x in 13 columns along with the total that confirms the 13 and not the 11?

A: We are sorry for the confusion. To clarify, the 11 variables refer to those common across all models, including both overall and seasonal scenarios. The 13 variables, as indicated in Table 1 (now S1 Table), represent the total number of variables considered for each scenario, with 11 of them being shared across all models.

We modified the manuscript according to reviewer and editor feedbacks (line 225). We hope this clears up the confusion. 

2. The authors list the 11 here as well as several that were not included in overall or seasonal models which, in turn, is presenting "results" in the Materials and Methods? There is also a section in the Results (Lines 365-368) that says out of the 20 variables that 13 were selected in the overall and seasonal suitability models that references Table 1. It is not clear why some of this analysis is in the Materials and Methods and some is in Results?

A: We understand the concern regarding the placement of certain details in the Materials and Methods and in the Results sections. To clarify, both sections refer to the same analysis, not to two different analyses. We simply decided to include a brief sentence in the Results section to describe the results of the selection of variables, as we aimed to present the process of variable choice for clarity, transparency and reproducibility. We acknowledge that this might seem repetitive. However, by placing and recalling this information in the Results, we hoped to maintain a logical flow of the modeling process, making it easier for readers to follow the progression of the analysis. We specified that the collinearity/multicollinearity analysis refers to the one described in material and methods section in line 362. 

We hope this explanation clarifies our approach. 

3. Lines 226-228 identifies several variables that were excluded in all scenarios (overall and seasonal, I presume), and interestingly, 2 of the variables (Temperature and NDVI) are those averaged over a 10 year span that I mentioned was not appropriate in my previous review (see below comments from my previous review and author response).

A: We thank the reviewer for providing additional arguments. We modified the manuscript according to the reviewer and editor’s suggestions. We believe this revision will streamline the manuscript and better align with the flow of the content (see below).

4. Table 1 repeats some of what is in text in both sections and I would still suggest that it be deleted. If the authors believe it has value, then why not include mean (SDs) for each variable during each season and overall instead of just an x? An entire page for a table with x’s that could be included in 4 sentences in the Materials and Methods (like Lines 226-228 so one line for overall and one line for each season) does not seem necessary and I don’t see how it helps the “flow” considering it is also presented in the Results?

A: We really appreciate reviewer effort in providing such detailed arguments. We thank the reviewer for the thoughtful comments and su

---

## [Editor Report · Decision Letter 2]

2 Jan 2025

Habitat suitability mapping and landscape connectivity analysis to predict African swine fever spread in wild boar populations: a focus on Northern Italy

PONE-D-24-33277R2

Dear Dr. Faustini,

We’re pleased to inform you that your manuscript has been judged scientifically suitable for publication and will be formally accepted for publication once it meets all outstanding technical requirements.

Kind regards,

Francesco Bisi, Ph.D.

Academic Editor

PLOS ONE
---

## [Editor Report · Acceptance letter]

22 Jan 2025

PONE-D-24-33277R2 

PLOS ONE

Dear Dr. Faustini, 

I'm pleased to inform you that your manuscript has been deemed suitable for publication in PLOS ONE. Congratulations! Your manuscript is now being handed over to our production team.

Kind regards, 

on behalf of

Dr. Francesco Bisi 

Academic Editor

PLOS ONE